physical chemistry/organic chemistry/
synthetic chemistry

vapour pressure, thermogravimetry,
electrochemical synthesis, characterization,
hafnium isopropoxide, hafnium *n*-propoxide

**Author for correspondence:**
Yongming Chen
e-mail: csuchenyongming@163.com

This article has been edited by the Royal Society of Chemistry, including the commissioning, peer review process and editorial aspects up to the point of acceptance.

# Determination of the vapour pressure curves and vaporization enthalpies of hafnium alkoxides using thermogravimetric analysis

## Changhong Wang, Shenghai Yang and Yongming Chen

School of Metallurgy and Environment, Central South University, Changsha 410083, People's Republic of China

CW, 0000-0002-5069-0877; SY, 0000-0003-2598-909X;
YC, 0000-0002-4929-8157

In order to identify a volatile metallo-organic precursor for the deposition of hafnium oxide ($HfO_2$) films for atomic layer deposition (ALD) applications, the evaporative properties of hafnium alkoxides (hafnium isopropoxide, hafnium *n*-propoxide and hafnium *n*-butoxide) were investigated using thermogravimetric analysis. These hafnium alkoxide samples were synthesized by the electrochemical method and characterized by Fourier transform infrared spectroscopy, nuclear magnetic resonance and inductively coupled plasma analysis techniques. The characterization results indicated that the products were 99.997% high-purity hafnium alkoxides and could meet the requirement of purity considering the usage of making $HfO_2$ gate oxide by ALD. Synthesized samples were subjected to a simultaneous thermogravimetric–differential thermal analysis unit at $10\,K\,min^{-1}$ in a dry nitrogen atmosphere flowing at $100\,ml\,min^{-1}$. Benzoic acid was used to calculate a calibration constant, which could then be inserted into a modified Langmuir equation to calculate vapour pressure curves for hafnium isopropoxide and hafnium *n*-propoxide. Detailed vapour pressure data for the $HfO_2$ precursor hafnium alkoxides were determined. The vapour pressure curve of hafnium isopropoxide was constructed within the first stage, and calculated to be $\ln p = 31.157\ (\pm 0.200) - 13130.57\ (\pm 56.50)/T$. Hafnium *n*-propoxide and hafnium *n*-butoxide were simultaneously undergoing evaporation and decomposition, thus making calculations invalid.

# 1. Introduction

Nowadays, thermogravimetry (TG) has been well established as a common and rapid technique to study the thermal property for volatile chemicals [1], especially their vapour pressure curves and vaporization enthalpies ($\Delta H_{vap}$). Using such a method, some studies have been carried out for the volatile chemicals: pharmaceuticals [2], UV absorbers [3], antioxidants [4], organometallic compounds [5] and biological agents [6]. In earlier years, TG was used in an isothermal condition, but more recently, an increasing temperature heating programme, which leads to faster data collection and reduces the number of samples required [7,8], has been used in TG.

Hafnium alkoxide is mainly used for the deposition of hafnium oxide ($HfO_2$) layers by atomic layer deposition (ALD) [9], and these $HfO_2$ layers are among the most promising high-*k* dielectric candidates to meet the standard for substituting the traditional $SiO_2$ gate oxide in semiconductor devices [10,11]. As a precursor must be thermally stable under the conditions that are required to transport its vapours to the substrate zone, to avoid pressure build-up and escape of material, thermal property is thought to be of prime consideration in evaluating the feasibility of a metallo-organic compound as a precursor in ALD [12]. Because vapour pressure (VP) is important to predict a desired condition and is favourable for process investigation, the VP can be thought to be a crucial parameter of thermal property for selecting a precursor suited to ALD [13]. VP curves can be expressed using the Antoine equation, but the corresponding data obtained by the Antoine equation are either available for a limited number of compounds or cover a limited temperature interval [14,15]. Therefore, TG seems a very promising method for the determination of the vapour pressure curves and vaporization enthalpies, because it can make us obtain data faster and acquire data in a greater quantity and in a larger temperature interval.

Although the most common method of preparing hafnium alkoxides is based on the halide synthesis, the direct electrochemical synthesis of hafnium alkoxides has a greater promise owing to its remarkable advantages. Detailed reasons were described previously in our studies [16–19]. Many metal alkoxides, such as Y, Ti, Nb, Ta, Mo, W, Cu, Ge and Sn, were produced by this technique [20,21]. In 1995, hafnium alkoxides were successfully obtained by Turevskaya *et al.* [22] with the electrochemical method involving electrolysis of an anhydrous alcohol solution containing tetrabutylammonium bromide with a platinum or stainless steel plate cathode and a hafnium anode. In our earlier works, we have also successfully prepared several niobium and tantalum alkoxides and hafnium ethoxide by electrochemical process, and some of them have been investigated by thermogravimetric analysis (TGA) methods [17,23]. In Yang *et al.*'s work [24], *t*-butoxide was measured by TGA method and was detected giving a residue of almost 30%. However, the characterization and thermal property analyses by TGA methods of the hafnium isopropoxide, hafnium *n*-propoxide and hafnium *n*-butoxide synthesized by electrochemical process have not been investigated so far.

In this paper, the hafnium alkoxides (hafnium isopropoxide, hafnium *n*-propoxide and hafnium *n*-butoxide) synthesized by electrochemical process are characterized by Fourier transform-infrared spectroscopy (FT-IR), $^1$hydrogen-nuclear magnetic resonance ($^1$H-NMR) and inductively coupled plasma (ICP) analysis techniques. Thermal properties and the vapour pressure curves for the hafnium alkoxides (hafnium isopropoxide, hafnium *n*-propoxide and hafnium *n*-butoxide) are investigated using TGA methods.

# 2. Material and methods

## 2.1. Materials and sample preparation

Anhydrous isopropanol (purity: 99.97%), *n*-propanol (purity: 99.97%) and *n*-butanol (purity: 99.97%) were purchased from Tianjin Hengxing Chemical Preparation Corporation. Tetraethylammonium bromide (Et₄NBr) was supplied by Sinopharm Chemical Reagent Corporation Limited. All reagents were used without further purification. The electrolytic cell with the dimensions of 16.0 cm (length) × 8 cm (width) × 23 cm (height) was adopted and made of polypropylene. Cathode adopted a stainless steel plate, whose working area was 18.0 cm × 13.0 cm. Anode adopted a hafnium plate (2.0 kg), whose working area was 17.0 cm × 12.0 cm. The hafnium plates were made from hafnium powders of metallurgical grade with self-resistance sintering, electron bombardment and rolling treatment. To determine its chemical composition, hafnium plate was dissolved in hydrofluoric acid of a required concentration. The impurity contents were measured by ICP-Mass Agilent 7500a analyzer, and the corresponding result is presented in table 1.

Electrochemical synthesis of the hafnium alkoxides was carried out under the fixed experimental conditions: anhydrous alcohol 2.2 l, conductive agent Et₄NBr 0.056 mol kg$^{-1}$, required solution temperature (the synthesis of hafnium isopropoxide, hafnium *n*-propoxide and hafnium *n*-butoxide

**Table 1.** Chemical compositions of hafnium anode (mass fraction, ppm).

| Zr | Cr | Fe | K | Mg |
|----|----|----|----|----|
| 15 | <3 | 5 | <2 | <7 |
| Na | Mo | Si | Ni | W |
| <5 | <4 | <3 | 2 | 3.6 |

was carried out at 355.55 K, 370.25 K and 390.85 K, respectively), polar distance 2 cm and applied current 2 A (current density 98 A m$^{-2}$). After electrochemical synthesis under this condition, the hafnium alkoxide and corresponding alcohol solution were distilled at ambient pressure to separate redundant alcohol at the required temperature. Then the distillation temperature was raised to the corresponding temperature (around 423.15 K) to remove a little amount of ester. Finally, the crude hafnium alkoxide solution was distilled at a pressure of 5 kPa and at a required oil bath temperature. The condensate (hafnium isopropoxide, hafnium *n*-propoxide or hafnium *n*-butoxide) was preserved in a dry nitrogen-sealed glass bottle to keep it from moisture.

## 2.2. Characterization

The FT-IR spectrum was recorded with a Nicolet Avatar 360 IR spectrometer operating in the region of 4000−400 cm$^{-1}$. The resolution and the scan times of the FT-IR measurements were 4 cm$^{-1}$ and 32, respectively. The $^1$H-NMR spectrum was measured with an Inova-400 (Varian) nuclear magnetic resonance spectrometer, and chloroform-d was adopted as the solvent. To determine the impurity contents of hafnium alkoxides, an amount of water was added into hafnium alkoxides solutions for the hydrolysis reaction (Hf(OR)$_4$ + 2H$_2$O → HfO$_2$ + 4ROH). After desiccation in a desiccator under vacuum conditions at 373.15 K for 12 h, the sample was calcined in a muffle furnace at 1073.15 K for 2 h. The impurity contents were obtained by ICP-Mass Agilent 7500a analyzer and the impurity content of hafnium oxide was converted into that of hafnium alkoxides.

## 2.3. Thermogravimetric analysis

Thermogravimetric measurements were carried out in a nitrogen atmosphere using SDT Q600 V8.0 Build 95 thermoanalytical equipment (flux rate 100 ml min$^{-1}$, heating rate 10 K min$^{-1}$, temperature interval 298−1073 K, sample mass 5−10 mg and surface area of the TG crucible 0.227 cm$^2$). Each sample was analysed at least three times to check reproducibility.

Vapour pressure curves can be expressed using the Antoine equation [25]:

$$\log p = \frac{A - B}{T + C},$$ (2.1)

where $p$ is the vapour pressure, $T$ is the absolute temperature, and $A$, $B$ and $C$ are the Antoine's constants in a required temperature range. However, the data obtained by the Antoine equation are either available for a limited number of compounds or cover a limited temperature interval.

The principle of using TG to estimate the vapour pressure is based on the Langmuir equation [26]:

$$\frac{1}{a}\frac{dm}{dt} = p\alpha\sqrt{\frac{M}{2\pi RT}},$$ (2.2)

where $dm/dt(a)$ is the rate of mass loss per unit area (kg s$^{-1}$ m$^{-2}$), $p$ the vapour pressure (Pa), $M$ the relative molecular mass of the vapour of the evaporating compound (kg mol$^{-1}$), $T$ the absolute temperature (K) and $\alpha$ the vaporization coefficient. Rearranging the Langmuir equation gives

$$p = kv,$$ (2.3)

where $k = \sqrt{2\pi R}/\alpha$ and $v = (1/a)(dm/dt)\sqrt{T/M}$. It has been demonstrated that $k$ is dependent on the instrument and independent of the sample [27]. However, the value of $v$ is independent of the instrument and dependent on the sample. To obtain the vapour pressure of a substance whose Antoine constants are not known, the significant procedure is to find a suitable calibration material that is known to be thermally stable within the required temperature range and that its Antoine constants are known. Knowing the vapour pressure of the standard compound at different temperatures and obtaining $dm/dt$ from TG/DTG plots can determine the value of $k$. Benzoic acid

has been assumed as a suitable material for this role [28,29]. The effect of temperature on the vapour pressure of benzoic acid can be expressed by equation (2.1), in which A−C are the Antoine constants taken in a certain temperature range for the evaporating species. For benzoic acid, the values of A−C in the temperature range of 405–523 K are 7.80991, 2776.12 and 43.978, respectively [29]. With the aid of equation (2.3), the linear relationship of $p$ and $v$ should be plotted, and the slope equals $k$. Once the value of $k$ is determined, vapour pressure for the other substances, whose Antoine constants are not known, can be calculated using equation (2.3). The vaporization enthalpy can be calculated from the vapour pressure−temperature data using the Clausius−Clapeyron equation:

$$\ln p = A - \frac{\Delta H}{RT}. \tag{2.4}$$

From this equation, a plot of $\ln p$ versus $1/T$ should give a straight line, the slope of which is $-\Delta H/R$. Thus, the enthalpy of vaporization ($\Delta H_{vap}$) can be calculated.

# 3. Results and discussion

## 3.1. Characterization

Figure 1 presents the FT-IR spectra of hafnium isopropoxide, hafnium $n$-propoxide and hafnium $n$-butoxide. The several peaks within the range 2974–2852 cm$^{-1}$ correspond to $\delta$(C—H) stretching vibration of alkoxy groups. Peaks at around 2970 cm$^{-1}$ and 2860 cm$^{-1}$ correspond to asymmetric and symmetric stretch vibrations, respectively. The evident bands at around 1440 cm$^{-1}$ and 1376 cm$^{-1}$ are due to the $\delta$(C—H) bend vibrations. Peaks at 1440 and 1462 cm$^{-1}$ correspond to scissoring vibration of methylene. The peak at around 1376 cm$^{-1}$ is ascribed to symmetric bending vibration of methyl groups. Bands in the range 1200–1000 cm$^{-1}$ are related to the C—O vibrations of alkoxy groups bound to Hf. The broad envelopes of bands below 630 cm$^{-1}$ are due to the Hf—O stretching modes appearing along with bending and torsional modes of the ligands [30]. No bands can be observed within the range 3600–3100 cm$^{-1}$ in figure 1, indicating that the samples were well preserved and the partial hydrolyzation has not occurred.

Nuclear magnetic resonance studies on hafnium alkoxides have been a very instrumental way to understand and analyse the structure of these materials. Figure 2 presents the $^1$H-NMR spectra of Hf(OPr$^i$)$_4$, Hf(OPr$^n$)$_4$ and Hf(OBu$^n$)$_4$ in CDCl$_3$ in regard to tetramethylsilane. Solvent peaks are clearly seen at 7.26 ppm. Peak assignments in figure 2a are as follows: the peaks around 4.45 ppm are attributed to —OCH— (H$_A$). The major signals around 1.23 ppm are due to two CH$_3$— (H$_B$), namely the H atom of a couple of terminal methyl. The proportion of spectral integral area is 4.00 : 24.21, which is in good agreement with the stoichiometry of the two kinds of ligands in the compound Hf(OCH(CH$_3$)$_2$)$_4$. The multiple split peaks of CH$_3$— and —CH$_2$— appear due to the H—H coupling effect between methyl and methylene [30]. Figure 2b shows the spectrum measured for Hf(O(CH$_2$)$_2$CH$_3$)$_4$. The peaks around 4.01 ppm are assigned to OCH$_2$— (H$_A$). Compared with Hf(OPr$^i$)$_4$, the peak of H atom in this position moves to lower field because of the enhanced shielding effect of the methyl branched chain [30]. The signals around 1.58 ppm and 0.9 ppm correspond to —CH$_2$— (H$_B$) and CH$_3$— (H$_C$), respectively. The proportion of spectral integral area is 8.00:8.08:12.24, agreeing well with the stoichiometry of the three kinds of ligands in the compound Hf(O(CH$_2$)$_2$CH$_3$)$_4$. Moreover, as shown in figure 2c, the peaks around 3.61 ppm, 1.56 ppm, 1.39 ppm and 0.95 ppm correspond to OCH$_2$— (H$_A$), —CH$_2$— (H$_B$), —CH$_2$— (H$_C$) and CH$_3$— (H$_D$), respectively. As electron-attracting effect of the oxygen atom is weak and it is difficult to have a large impact on the distribution of electron cloud, the chemical shifts of H$_B$ and H$_C$ are very contiguous. As they are both normal alkoxides, the chemical shift of terminal CH$_3$— (H$_D$ in figure 2c) equals that of Hf(OPr$^n$)$_4$. The proportion of spectral integral area is 8.00:8.24:8.00:11.84, which is in accord with the stoichiometry of the four kinds of ligands in the compound Hf(O(CH$_2$)$_3$CH$_3$)$_4$. The normal hafnium alkoxides were not monomers but trimers with a triangular cluster structure [22]. In addition, it should be noted that while the samples were well preserved and the partial hydrolyzation had not occurred before $^1$H-NMR measurement, there are some impure peaks in NMR images. The impurity peaks in the spectra may be attributed to the hydrolysis of Hf(OR)$_4$ by the trace amount of water in CDCl$_3$ solvent, because the Hf(OR)$_4$ is acutely sensitive to water and easily suffers hydrolysis.

Table 2 presents the contents of impurities in hafnium alkoxide. The contents of Zr in hafnium alkoxides are between 10.4 and 11.2 ppm, and other impurities are between 1.1 and 3.2 ppm.

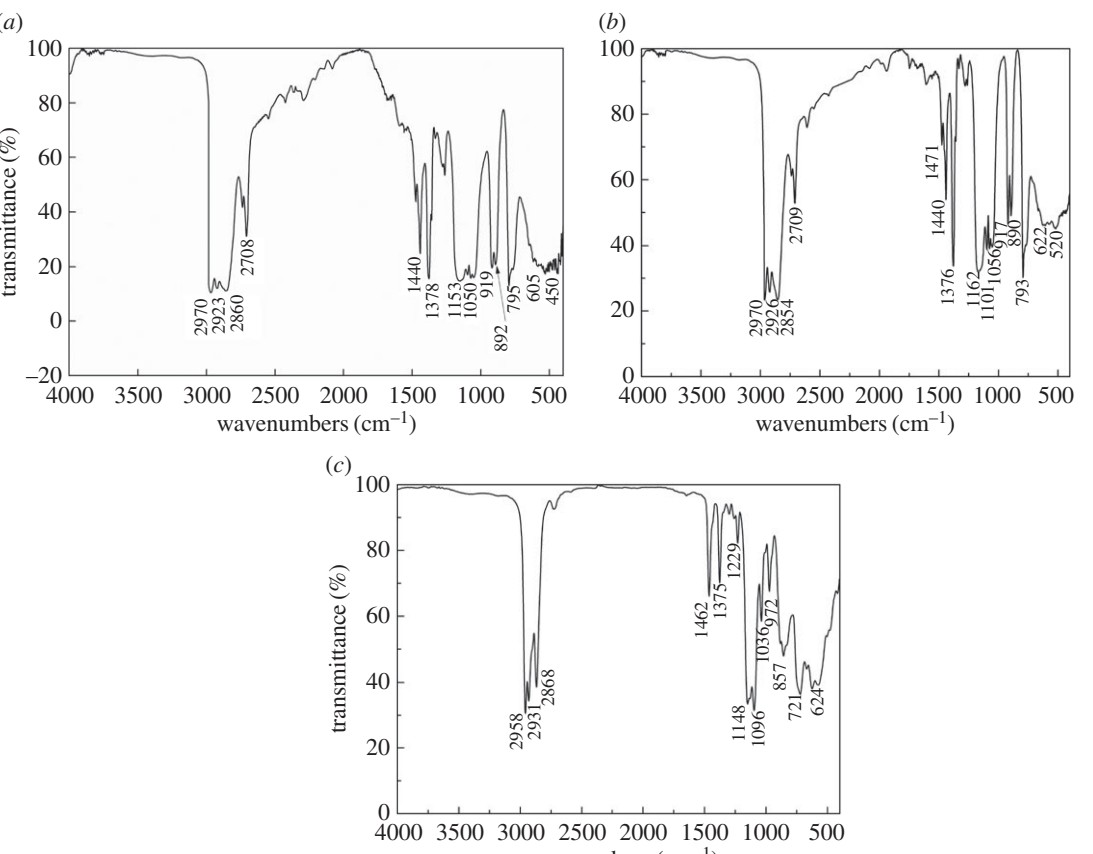

**Figure 1.** FT-IR spectra of hafnium alkoxide samples: (*a*) hafnium isopropoxide; (*b*) hafnium *n*-propoxide; (*c*) hafnium *n*-butoxide.

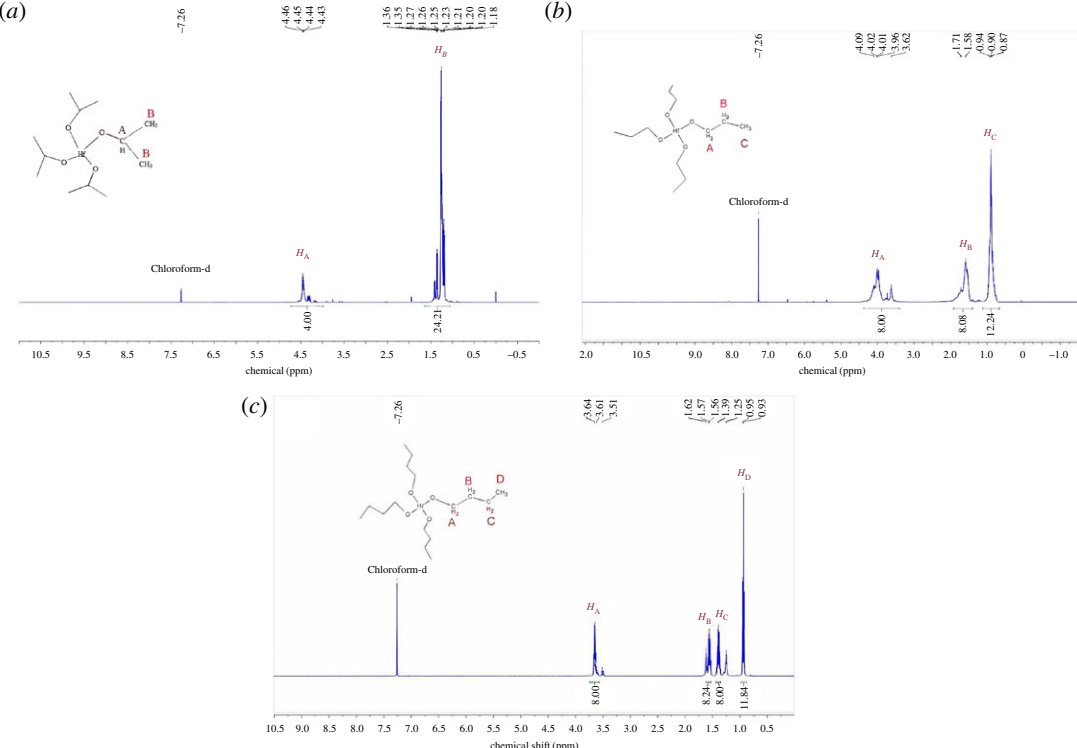

**Figure 2.** ¹H-NMR spectra of hafnium alkoxide samples: (*a*) hafnium isopropoxide; (*b*) hafnium *n*-propoxide; (*c*) hafnium *n*-butoxide.

**Table 2.** Impurity contents in hafnium alkoxides (ppm).

| | Zr | Hf | Al | Li | Mg | Fe | Cu | W | U | Th | Ta | Si | P |
|---|---|---|---|---|---|---|---|---|---|---|---|---|---|
| Hf(OPr$^i$)$_4$ | 10.4 | — | 1.4 | 1.6 | 1.5 | 1.9 | 1.5 | 1.7 | 1.5 | 1.4 | 2.9 | 1.2 | 1.3 |
| Hf(OPr$^n$)$_4$ | 10.8 | — | 1.8 | 1.9 | 1.8 | 2.1 | 1.7 | 2.3 | 1.8 | 1.8 | 3.2 | 2.1 | 2.1 |
| Hf(OBu$^n$)$_4$ | 11.2 | — | 1.6 | 2.1 | 1.1 | 2.3 | 1.2 | 2.1 | 1.5 | 1.1 | 2.6 | 2.2 | 1.6 |

Therefore, the purity of hafnium alkoxides by reduced pressure distillation can be up to 99.997%, and can meet the requirement of purity considering the usage of making HfO$_2$ gate oxide by ALD. The content of Zr is relatively high, and causes less harm to the quality than other elements.

## 3.2. Thermal properties analysis and vapour pressure estimation

### 3.2.1. Hafnium isopropoxide

Thermal property is of prime consideration in evaluating the feasibility of a metallo-organic compound as a precursor in ALD, demonstrating the conditions required to transport material from its source container to the deposition zone.

A typical thermogravimetric–differential thermal analysis (TG–DTA)/DTG plot for hafnium isopropoxide acquired at a heating rate of 10 K min$^{-1}$ is presented in figure 3$a$. Below 661 K, it is clear that the DTA curve has two endothermic peaks, the first of which is considered to be attributed to the evaporation of hafnium isopropoxide between 526 and 631 K, and the second is probably assigned to the pyrolytic decomposition of hafnium isopropoxide between 631 and 659 K. This result agrees well with Bradley *et al.*'s study [21]. They found that the hafnium compound began to decompose at a very low rate above 525 K. However, the sample suddenly starts to decompose significantly at 626 K. From its physical appearance at room temperature, hafnium isopropoxide synthesized seems a viscous liquid, which can explain why no melting peak is seen in the DTA curve presented for Hf(OPr$^i$)$_4$ synthesized in this work. It can be observed in the TG curve that as the temperature is increased from 298 K to 500 K, the mass loss rate gradually increases to 5.83%. This result is mainly due to the evaporation of a small amount of hafnium isopropoxide as a result of its lower saturated vapour pressure at lower temperature. Afterwards, the TG curve exhibits a comparatively rapid mass loss between 500 and 589 K. On the DTG curve, a rapid rate of mass loss is observed after 508 K. The rate of mass loss starts to increase until reaching the maximum evaporation rate at the DTG peak temperature of 574 K. A white residue (HfO$_2$) of about 22% of the total mass loss remained in the sample crucible at the end of the experiment, indicating that a part of pyrolytic decomposition of hafnium isopropoxide had occurred.

The TG/DTG plot for benzoic acid is shown in figure 3$b$, from which $v$ at different $T$ can be determined according to equation (2.3). In the DTA plot for benzoic acid, there was one endothermic peak in a lower temperature region, which is taken to be the melting point ($T_m$), and one endothermic peak in the upper temperature region, which is taken to represent evaporation [29]. Using the Antoine equation, equation (2.1) of benzoic acid in the evaporation process, the vapour pressure ($p$) can be obtained as a function of temperature. Then the dependence of $p$ on $v$ is plotted, as presented in figure 4, from which the slope ($k$) of the fitting line is calculated to be $146\,533 \pm 823$ J$^{0.5}$ K$^{-0.5}$ mol$^{-0.5}$. The standard deviation was calculated from three determinations carried out under identical experimental conditions. Corresponding data, which have been used to construct figure 4, to calculate $k$ are given in table 3. The values of $p_{calc.}$ in table 3 are obtained from the fitting line of plot of $p$ against $v$. Based on the value of $k$, the vapour pressure–temperature ($p$-$T$) curve for hafnium isopropoxide is calculated and plotted, as shown in figure 5. Through the $p$-$T$ data, a plot of ln$p$ against $1/T$ for hafnium isopropoxide is made and gives a straight fitting line, as illustrated in figure 6, from which the slope gives $-\Delta H_{vap}/R$ according to equation (2.4) and hence the vaporization enthalpy ($\Delta H_{vap}$) can be calculated ($109.2 \pm 0.47$ kJ mol$^{-1}$). The standard deviation was calculated from three determinations carried out under identical experimental conditions. This compares well with a value of $106 \pm 1.00$ kJ mol$^{-1}$, taken from a table of enthalpy of evaporation against temperature in Bradley *et al.*'s study [21]. Values of $p$, ln$p$, ln$p_{calc.}$ and $T^{-1}$ for hafnium isopropoxide at the temperature ranging from 485 K to 573 K, which have been used to construct figures 5 and 6, are recorded in table 4, in which values of ln$p_{calc.}$ are obtained from the fitting line of

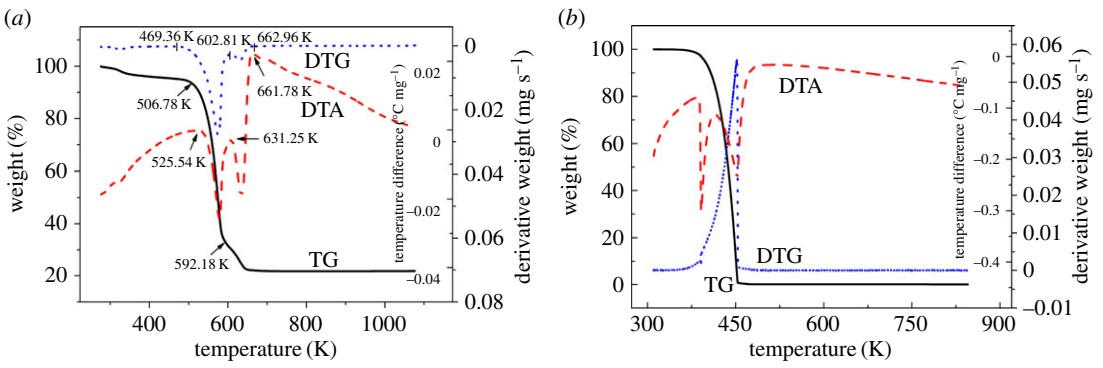

**Figure 3.** TG–DTA/DTG plots for (a) hafnium isopropoxide and (b) benzoic acid (10 K min$^{-1}$, 100 ml min$^{-1}$, N$_2$).

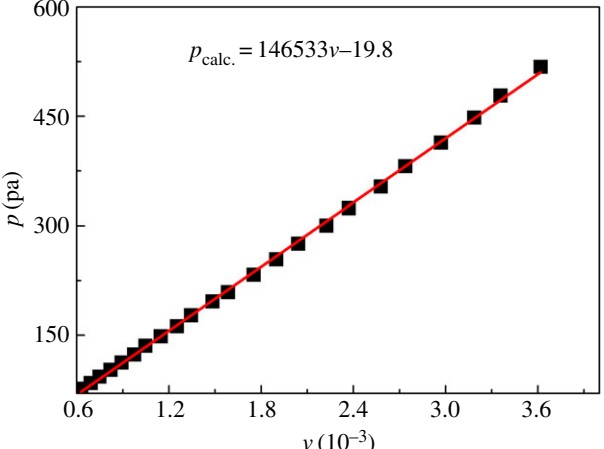

$$p_{calc.} = 146533v - 19.8$$

**Figure 4.** The determination of benzoic acid independent variable (k) for the evaporation process (10 K min$^{-1}$, 100 ml min$^{-1}$, N$_2$).

**Table 3.** Corresponding data to calculate k. The data in table 3 correspond to those in figure 4. The reported values are averages of three determination results carried out under identical experimental conditions. Standard uncertainties are u(v) = ±0.000001, u($p_{calc.}$) = ±1.4 Pa.

| $v/10^{-3}$ | $p/Pa^{29}$ | $p_{calc.}/Pa$ | $v/10^{-3}$ | $p/Pa^{29}$ | $p_{calc.}/Pa$ |
|---|---|---|---|---|---|
| 0.63 | 76.4 | 72.5 | 1.751 | 233.1 | 236.8 |
| 0.691 | 84.3 | 81.4 | 1.898 | 254.2 | 258.3 |
| 0.748 | 93 | 89.8 | 2.041 | 275.6 | 279.3 |
| 0.82 | 102.4 | 100.3 | 2.224 | 300.3 | 306.1 |
| 0.892 | 112.4 | 110.9 | 2.37 | 324.2 | 327.5 |
| 0.974 | 123.6 | 122.9 | 2.578 | 353.7 | 358 |
| 1.047 | 135.6 | 133.6 | 2.737 | 381.5 | 381.3 |
| 1.147 | 148.5 | 148.3 | 2.97 | 413.8 | 415.4 |
| 1.252 | 162.3 | 163.6 | 3.186 | 448.3 | 447 |
| 1.344 | 177.3 | 177.1 | 3.357 | 478.4 | 472.1 |
| 1.483 | 196.3 | 197.5 | 3.618 | 517.9 | 510.3 |
| 1.584 | 209.1 | 212.3 | | | |

plot of ln$p$ against $1/T$. The 45 points from 485 K to 573 K, used to compute equation (2.4), are representative of a large number of measurement data, giving a standard deviation of ± 0.035 in Δln$p$ (ln$p$ − ln$p_{calc.}$). Parameters of the fitting line ($y = a + bx$) for hafnium isopropoxide in the plot of ln$p$ against $1/T$, figure 6, are listed in table 5. The standard errors of intercept and slope for the fitting

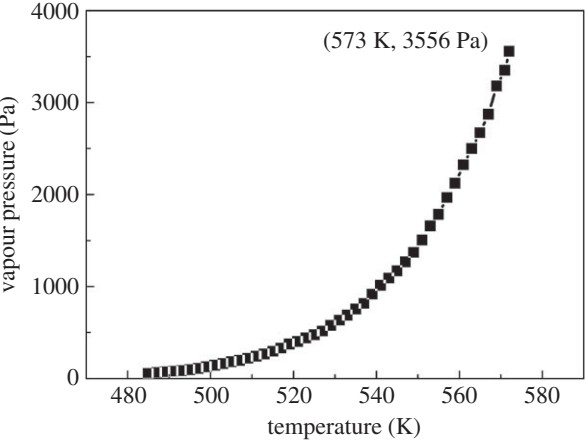

**Figure 5.** Vapour pressure-temperature ($p$-$T$) curve of hafnium isopropoxide.

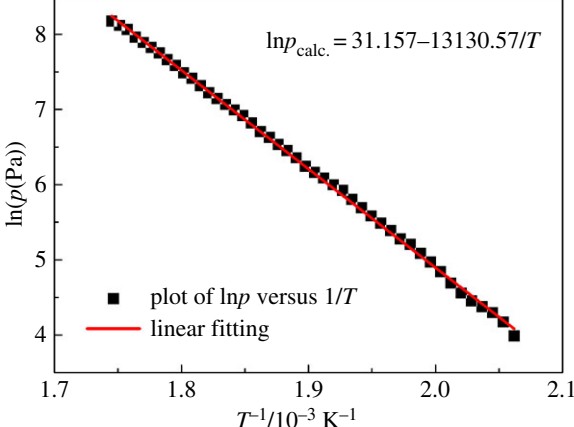

**Figure 6.** Plot of ln$p$ versus $1/T$ for hafnium isopropoxide.

line are 0.34% and 0.43%, respectively, confirming the validity of the fitting equation, and indicating that the fitting equation may be extrapolated by these points with confidence. The influence of various heating rates (2, 5, 10, 15 and 20 K min$^{-1}$) at a constant nitrogen flow rate of 100 ml min$^{-1}$ on the $\Delta H_{vap}$ was investigated, and the results are recorded in table 6. Similarly, the influence of various nitrogen flow rates (50, 100, 150, 200 and 250 ml min$^{-1}$) was studied at a constant heating rate of 10 K min$^{-1}$ on the $\Delta H_{vap}$, and the results are listed in table 7. It is observed that with increasing heating rate and gas flow rate, corresponding vaporization enthalpy of hafnium isopropoxide has a small decrease. The presence of 22% white residues in the crucible at the end of the TG experiment was an indication that a part of pyrolytic decomposition of hafnium isopropoxide occurred in this temperature range. However, because trace white residue caused by the decomposition of hafnium isopropoxide appeared on the TG crucible at the end of 485–573 K, it can be deduced that the decomposition is likely to be much slower at the lower temperatures and, therefore, using the data for calculating vapour pressures of the compound is valid when the temperature interval is between 485 and 573 K [31].

### 3.2.2. Hafnium $n$-propoxide

A typical TG–DTA/DTG plot of hafnium $n$-propoxide at a heating rate of 10 K min$^{-1}$ at a nitrogen flowing rate of 100 ml min$^{-1}$ is shown in figure 7. On the DTA plot, two endothermic peaks are clearly observed, which are due to the evaporation and/or decomposition of hafnium $n$-propoxide in the temperature ranges 560–631 K and 631–677 K. The TG curve shows that when the temperature is increased from 298 K to 551 K, the mass loss rate increases to 12.10%. This result is mainly due to the evaporation of a small amount of hafnium $n$-propoxide as a result of its lower saturated vapour

**Table 4.** Values of $p$/Pa, $\ln(p$/Pa$)$, $\ln(p_{calc.}$/Pa$)$ and $T^{-1}/10^{-3}$ (K$^{-1}$) for hafnium isopropoxide vapour at the temperature ($T$) ranging from 485 K to 573 K. The reported values are averages, of three determination results carried out under identical experimental conditions. Standard uncertainties are $u(T) = \pm0.05$ K, $u(p) = \pm1.0$ Pa, $u(p_{calc.}) = \pm0.9$ Pa.

| $T$/K | $T^{-1}/10^{-3}$ (K$^{-1}$) | $p$/Pa | $\ln(p$/Pa$)$ | $\ln(p_{calc.}$/Pa$)$ | $T$/K | $T^{-1}/10^{-3}$ (K$^{-1}$) | $p$/Pa | $\ln(p$/Pa$)$ | $\ln(p_{calc.}$/Pa$)$ |
|---|---|---|---|---|---|---|---|---|---|
| 485 | 2.062 | 53.9 | 3.99 | 4.08 | 531 | 1.883 | 635.5 | 6.45 | 6.43 |
| 487 | 2.053 | 64.9 | 4.17 | 4.2 | 533 | 1.876 | 688.3 | 6.53 | 6.52 |
| 489 | 2.045 | 73.5 | 4.3 | 4.3 | 535 | 1.869 | 755.4 | 6.63 | 6.62 |
| 491 | 2.037 | 79.6 | 4.38 | 4.41 | 537 | 1.862 | 815.8 | 6.7 | 6.71 |
| 493 | 2.028 | 86 | 4.45 | 4.53 | 539 | 1.855 | 916.3 | 6.82 | 6.8 |
| 495 | 2.02 | 95.4 | 4.56 | 4.63 | 541 | 1.848 | 1011.9 | 6.92 | 6.89 |
| 497 | 2.012 | 108.8 | 4.69 | 4.74 | 543 | 1.842 | 1090 | 6.99 | 6.97 |
| 499 | 2.004 | 126.7 | 4.84 | 4.84 | 545 | 1.835 | 1170.7 | 7.07 | 7.06 |
| 501 | 1.996 | 144.2 | 4.97 | 4.95 | 547 | 1.828 | 1267.3 | 7.14 | 7.15 |
| 503 | 1.988 | 161.6 | 5.09 | 5.05 | 549 | 1.821 | 1371.5 | 7.22 | 7.25 |
| 505 | 1.98 | 182.6 | 5.21 | 5.16 | 551 | 1.815 | 1503.6 | 7.32 | 7.33 |
| 507 | 1.972 | 196.1 | 5.28 | 5.26 | 553 | 1.808 | 1659 | 7.41 | 7.42 |
| 509 | 1.965 | 218.6 | 5.39 | 5.36 | 555 | 1.802 | 1784.1 | 7.49 | 7.5 |
| 511 | 1.957 | 241.3 | 5.49 | 5.46 | 557 | 1.795 | 1966.3 | 7.58 | 7.59 |
| 513 | 1.949 | 265.6 | 5.58 | 5.57 | 559 | 1.789 | 2123.6 | 7.66 | 7.67 |
| 515 | 1.942 | 296.1 | 5.69 | 5.66 | 561 | 1.783 | 2322.4 | 7.75 | 7.75 |
| 517 | 1.934 | 331.2 | 5.8 | 5.76 | 563 | 1.776 | 2498.8 | 7.82 | 7.84 |
| 519 | 1.927 | 373.6 | 5.92 | 5.85 | 565 | 1.77 | 2673.3 | 7.89 | 7.92 |
| 521 | 1.919 | 402.3 | 6 | 5.96 | 567 | 1.764 | 2872.8 | 7.96 | 7.99 |
| 523 | 1.912 | 440.2 | 6.09 | 6.05 | 569 | 1.757 | 3179.5 | 8.06 | 8.09 |
| 525 | 1.905 | 474.7 | 6.16 | 6.14 | 571 | 1.751 | 3350.1 | 8.12 | 8.17 |
| 527 | 1.898 | 515.6 | 6.25 | 6.24 | 573 | 1.745 | 3556.1 | 8.18 | 8.24 |
| 529 | 1.89 | 575.8 | 6.36 | 6.34 | | | | | |

**Table 5.** Parameters of the fitting line ($y = a + bx$) for hafnium isopropoxide in the plot of $\ln p$ against $1/T$. The data in table 5 correspond to those in figure 6. The reported values are averages of three determination results carried out under identical experimental conditions. Standard uncertainties are $u(a) = \pm0.200$, $u(b) = \pm56.50$.

| | value | standard error |
|---|---|---|
| intercept (a) | 31.157 | 0.107 |
| slope (b) | −13130.57 | 56.25 |

pressure at lower temperature, and/or the decomposition of a small part of hafnium $n$-propoxide. On the DTG curve, the mass loss rate evidently increases at 555 K and accelerates to the maximum value at 595 K. A white residue (HfO$_2$) of about 44% of the total mass loss remained in the sample crucible at the end of the experiment, indicating that a major amount of pyrolytic decomposition of hafnium $n$-propoxide had occurred.

Similarly, the $k$-value 146 533 J$^{0.5}$ K$^{-0.5}$ mol$^{-0.5}$ from the benzoic acid calculations is used to calculate the vapour pressure values for hafnium $n$-propoxide, and the related plot is shown in figure 8. Using the $p$-$T$ data, the plot of $\ln p$ versus $1/T$, as presented in figure 9, can be constructed. The linear feature allows $\Delta H_{vap}$ to be obtained from the slope, and it is calculated to be $169.4 \pm 2.23$ kJ mol$^{-1}$. Values of $p$, $\ln p$, $\ln p_{calc.}$ and $T^{-1}$ for hafnium $n$-propoxide at temperatures ranging from 543 K to 589 K, which have been used to construct figures 8 and 9, are recorded in table 8. The 24 points from 543 K to 589 K

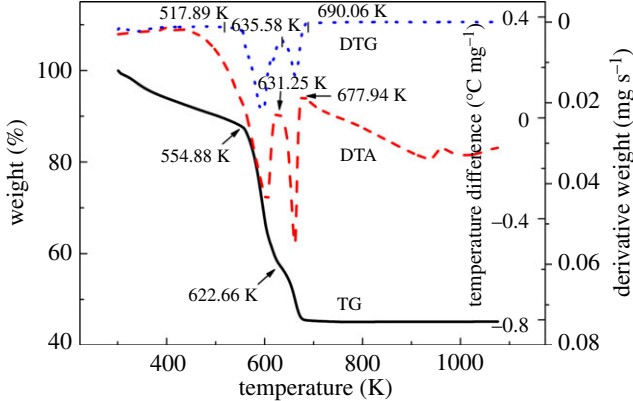

**Figure 7.** TG−DTA /DTG plots for hafnium $n$-propoxide (10 K min$^{-1}$, 100 ml min$^{-1}$, N$_2$).

**Table 6.** Effect of different heating rates on the vaporization enthalpies of hafnium isopropoxide at a constant nitrogen flow rate of 100 ml min$^{-1}$.

| $\beta$ (K min$^{-1}$) | $\Delta H_{vap}$ (kJ mol$^{-1}$) |
| --- | --- |
| 2 | 110.0 ± 0.48 |
| 5 | 109.9 ± 0.49 |
| 10 | 109.2 ± 0.47 |
| 15 | 108.9 ± 0.45 |
| 20 | 108.0 ± 0.46 |

**Table 7.** Effect of different nitrogen flow rates on the vaporization enthalpies of hafnium isopropoxide at a constant heating rate of 10 K min$^{-1}$.

| flow rate (ml min$^{-1}$) | $\Delta H_{vap}$ (kJ mol$^{-1}$) |
| --- | --- |
| 50 | 110.0 ± 0.46 |
| 100 | 109.2 ± 0.47 |
| 150 | 109.0 ± 0.49 |
| 200 | 108.4 ± 0.48 |
| 250 | 108.3 ± 0.47 |

used to compute equation (2.4) are representative of a large number of measurement data, giving a standard deviation of ±0.17 in $\Delta \ln p$ ($\ln p - \ln p_{calc.}$). Parameters of the fitting line ($y = a + bx$) for hafnium $n$-propoxide in the plot of $\ln p$ against $1/T$, figure 9, are listed in table 9. Compared with 0.34% and 0.43% from hafnium isopropoxide, the standard errors of intercept and slope for the fitting line are 3.59% and 4.07%, respectively, indicating the invalidity of the fitting equation for hafnium $n$-propoxide. A white residue of about 44% of the total mass loss remained in the sample crucible at the end of the experiment, and the amount of HfO$_2$ that would form if the compound in its entirety would turn into HfO$_2$ is 50.7%. As the vapour pressure values were calculated between 543 and 589 K, and because a major amount of white residue appeared on the TG crucible in the heating process between 543 and 589 K mass loss, it can be deduced that hafnium $n$-propoxide is simultaneously undergoing evaporation and significant decomposition and, therefore, using the data for calculating vapour pressures of the compound is invalid.

### 3.2.3. Hafnium $n$-butoxide

Hafnium $n$-butoxide was heated under the identical condition applied to the other alkoxides. Similarly, the DTA curve showed two endothermic peaks between 575–650 K and 663–691 K, both caused by the

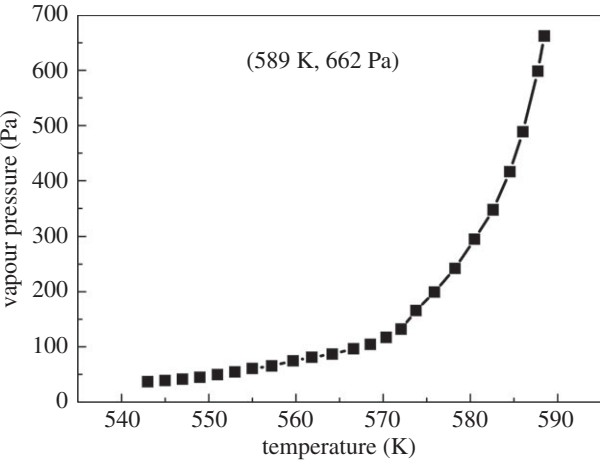

**Figure 8.** Vapour pressure-temperature curve of hafnium *n*-propoxide.

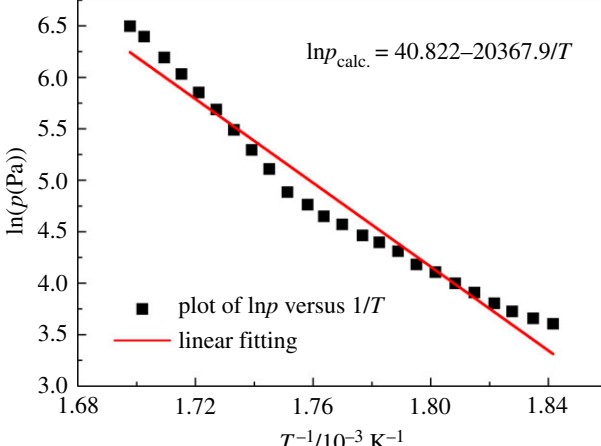

**Figure 9.** Plot of ln*p* versus 1/*T* for hafnium *n*-propoxide.

**Table 8.** Values of $p$/Pa, ln($p$/Pa), ln($p_{calc.}$/Pa) and $T^{-1}/10^{-3}$ (K$^{-1}$) for hafnium *n*-propoxide vapour at the temperature ($T$) ranging from 543 K to 589 K. The reported values are averages of three determination results carried out under identical experimental conditions. Standard uncertainties are u($T$) = $\pm$0.05 K, u($p$) = $\pm$0.8 Pa, u($p_{calc.}$) = $\pm$0.5 Pa.

| $T$/K | $T^{-1}/10^{-3}$ (K$^{-1}$) | $p$/Pa | ln($p$/Pa) | ln($p_{calc.}$/Pa) | $T$/K | $T^{-1}/10^{-3}$ (K$^{-1}$) | $p$/Pa | ln($p$/Pa) | ln($p_{calc.}$/Pa) |
|---|---|---|---|---|---|---|---|---|---|
| 543 | 1.842 | 36.8 | 3.61 | 3.31 | 567 | 1.764 | 104.5 | 4.65 | 4.89 |
| 545 | 1.835 | 38.8 | 3.66 | 3.45 | 569 | 1.757 | 117.1 | 4.76 | 5.04 |
| 547 | 1.828 | 41.5 | 3.73 | 3.59 | 571 | 1.751 | 132.1 | 4.88 | 5.16 |
| 549 | 1.821 | 44.9 | 3.8 | 3.73 | 573 | 1.745 | 165.6 | 5.11 | 5.28 |
| 551 | 1.815 | 49.8 | 3.91 | 3.86 | 575 | 1.739 | 199.2 | 5.29 | 5.4 |
| 553 | 1.808 | 54.5 | 4 | 4 | 577 | 1.733 | 242 | 5.49 | 5.53 |
| 555 | 1.802 | 60.8 | 4.11 | 4.12 | 579 | 1.727 | 295 | 5.69 | 5.65 |
| 557 | 1.795 | 65.4 | 4.18 | 4.26 | 581 | 1.721 | 347.9 | 5.85 | 5.77 |
| 559 | 1.789 | 74.5 | 4.31 | 4.39 | 583 | 1.715 | 416.3 | 6.03 | 5.89 |
| 561 | 1.783 | 81.1 | 4.4 | 4.51 | 585 | 1.709 | 488.7 | 6.19 | 6.01 |
| 563 | 1.776 | 86.8 | 4.46 | 4.65 | 587 | 1.704 | 598.6 | 6.4 | 6.12 |
| 565 | 1.77 | 96.6 | 4.57 | 4.77 | 589 | 1.698 | 662.2 | 6.5 | 6.24 |

**Table 9.** Parameters of the fitting line ($y = a + bx$) for hafnium $n$-propoxide in the plot of $\ln p$ against $1/T$. The data in table 9 correspond to those in figure 9. The reported values are averages of three determination results carried out under identical experimental conditions. Standard uncertainties are u(a) $= \pm 0.220$, u(b) $= \pm 268.5$.

|  | value | s.e. |
|---|---|---|
| intercept (a) | 40.822 | 1.467 |
| slope (b) | $-20367.9$ | 829.4 |

evaporation and/or decomposition of hafnium $n$-butoxide. The TG curve presented a two-stage mass loss between 575–643 K and 651–689 K. A white residue ($HfO_2$) of about 41.6% of the total mass loss remained in the sample crucible at the end of the experiment, and the amount of $HfO_2$ that would form if the compound in its entirety would turn into $HfO_2$ is 44.7%. As a major amount of white residue appeared on the TG crucible in the heating process between 575 and 623 K, it was deduced that hafnium $n$-butoxide was simultaneously undergoing evaporation and significant decomposition. Therefore, it was not possible to measure the vapour pressure curve of this material.

## 4. Conclusion

In this study, hafnium isopropoxide, hafnium $n$-propoxide and hafnium $n$-butoxide were prepared by electrochemical synthesis and subjected to characterization and TG–DTA. The characterization results demonstrate that these samples are high-purity hafnium alkoxides. The purity of hafnium alkoxides purified by reduced pressure distillation can be up to 99.997% and can meet the requirement of purity considering the usage of making $HfO_2$ gate oxide by ALD. TG–DTA results indicate that the vapour pressure curve of hafnium isopropoxide is constructed within the first stage, and calculated to be $\ln p = 31.157(\pm 0.200) - 13\,130.57(\pm 56.50)/T$. Hafnium $n$-propoxide and hafnium $n$-butoxide are simultaneously undergoing evaporation and decomposition, so calculations are rendered invalid.

Data accessibility. All the data are provided in the manuscript in the form of tables and figures, as shown in tables 1–9 and figures 1–9.

Authors' contributions. C.W. contributed to the design of the experiment as well as the acquisition and analysis of the data. All authors contributed to drafting the manuscript. S.Y. contributed to revising the article critically for important intellectual content.

Competing interests. We declare we have no competing interests.

Funding. The National Natural Science Foundation of China (no. 51374254) and the Fundamental Research Funds for the Central Universities of Central South University provided financial support for this work.

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
