## [Reviewer comments · Royal Society Open Science]

Review History

RSOS-180688.R0 (Original submission)

Review form: Reviewer 1

Is the manuscript scientifically sound in its present form?

No

Are the interpretations and conclusions justified by the results?

No

Is the language acceptable?

No

Is it clear how to access all supporting data?

Not Applicable

Do you have any ethical concerns with this paper?

No

Have you any concerns about statistical analyses in this paper?

Yes

Recommendation?

Major revision is needed (please make suggestions in comments)

Comments to the Author(s)

Please look in the attached .pdf file (Appendix A).

Review form: Reviewer 2 (Markku Leskelä)

Is the manuscript scientifically sound in its present form?

Yes

Are the interpretations and conclusions justified by the results?

Yes

Is the language acceptable?

Yes

Is it clear how to access all supporting data?

Yes

Do you have any ethical concerns with this paper?

No

Have you any concerns about statistical analyses in this paper?

No

Recommendation?

Accept with minor revision (please list in comments)

Comments to the Author(s)

My comments are:

Introduction 2nd paragraph: Hafnium alkoxide is mainly used in ALD [10]. This not true and the ref. 10 does not say that. Ref. 10 tells about the density of the films. A better reference what precursors have been used in ALD (and also in case of Hf) is J. Appl. Phys. 113 (2013) 021301.

The reference 11 is also bad if a general reference is wanted to show the importance of HfO₂ as high-k material. There are more than 5 000 papers on this topic in the literature and best is to refer to a review. A good example could be Mater. Sci. Reports 72 (2011) 97. Just to your information, in industrial scale ALD HfO_x are made for transistors from HfCl₄.

Same paragraph: ... vapor pressure is a crucial parameter for selecting precursors for ALD. Yes, volatility is the most important property. However, for practical use of precursors we need the

knowledge at which temperature we can reach about 0.1 mbar pressure. Pressure is not a critical parameter in an ideal ALD process – the only thing is that we have enough precursor molecules to saturate the surface. Having said that it does not mean that knowing the vapor pressure behavior is not important but it can be more important outside ALD.

The experimental part is OK. The measurements are made carefully and the knowledge on isopropoxide is useful. It is also useful to see that the other alkoxides decompose.

Some formal comments:

Tables 5 and 9 are very short. Is it necessary to present the numbers in a table.

Numbers in Tables 6 and 7 are almost constant. Are the tables needed?

Text for Table 3 is very short and not understandable in present form.

Review form: Reviewer 3

Is the manuscript scientifically sound in its present form?

Yes

Are the interpretations and conclusions justified by the results?

Yes

Is the language acceptable?

Yes

Is it clear how to access all supporting data?

Yes

Do you have any ethical concerns with this paper?

No

Have you any concerns about statistical analyses in this paper?

No

Recommendation?

Major revision is needed (please make suggestions in comments)

Comments to the Author(s)

This manuscript by Chen et al described the synthesis, characterization of two Hf alkoxides, and evaluated their potential in atomic layer deposition applications. The results and discussions are well organized. The manuscript is well written. However, due to the following reasons, major revision is recommended:

1. The authors claimed that they synthesized the Hf(OR)₄ compounds: 'the purity of hafnium alkoxides purified by reduced pressure distillation can be up to 99.997%.' It seems they did the calculation based on trace metal impurity. However, they didn't count organic impurities. If one checks the NMR spectra carefully (Figures 2a and 2b), there are obvious impure peaks that are adjacent to each product peak. This reviewer suspects there is unreacted alcohol. Did the authors do Elemental Analysis to compare with calculated C/H amounts? The authors likely used regular

CDCl₃ as deuterated solvent instead of anhydrous CDCl₃. If the authors obtained high purity product, the impurity such as trace water may lead to decomposition and thus impurity peaks. The purity can't be so high as 99.997% if considering the impurities by NMR.

2. The authors mentioned previous work by Yang et al. on Hf tetrabutoxide. Please compare the results with Yang's and comment.
3. Figure 2a: the integrals are not correct and should be 4H and 24H.
4. Figure 2b: the integrals are not correct and should be 8H, 8H and 12H.

Decision letter (RSOS-180688.R0)

18-Jun-2018

Dear Dr Wang:

Manuscript ID: RSOS-180688

Title: "Determination of the vapor pressure curves and vaporization enthalpies of hafnium alkoxides using thermogravimetric analysis"

Thank you for submitting the above manuscript to Royal Society Open Science. Your paper was sent to reviewers and their comments are included at the bottom of this letter.

In view of the concerns raised by the reviewers, the manuscript has been rejected in its current form. However, a new manuscript may be submitted which takes into consideration these comments.

Please note that resubmitting your manuscript does not guarantee eventual acceptance, and that your resubmission will be subject to peer review before a decision is made.

Your resubmitted manuscript should be submitted by 16-Dec-2018. If you are unable to submit by this date please contact the Editorial Office.

Yours sincerely,
Dr Laura Smith, MRSC
Publishing Editor, Journals
Royal Society of Chemistry,
Thomas Graham House,
Science Park, Milton Road,
Cambridge, CB4 0WF, UK

Royal Society Open Science - Chemistry Editorial Office

On behalf of the Subject Editor Professor Anthony Stace and the Associate Editor Dr Hazel Cox

REVIEWER(S) REPORTS:

Associate Editor Comments to Author ():

RSC Associate Editor:

Comments to the Author:

All of the reviewers comments need to be considered and addressed in detail before it can be reconsidered for publication.

RSC Subject Editor:

Comments to the Author:

(There are no comments.)

Reviewers' Comments to Author:

Reviewer: 1

Comments to the Author(s)

Please look in the attached .pdf file.

Reviewer: 2

Comments to the Author(s)

My comments are:

Introduction 2nd paragraph: Hafnium alkoxide is mainly used in ALD [10]. This not true and the ref. 10 does not say that. Ref. 10 tells about the density of the films. A better reference what precursors have been used in ALD (and also in case of Hf) is J. Appl. Phys. 113 (2013) 021301.

The reference 11 is also bad if a general reference is wanted to show the importance of HfO₂ as high-k material. There are more than 5 000 papers on this topic in the literature and best is to refer to a review. A good example could be Mater. Sci. Reports 72 (2011) 97. Just to your information, in industrial scale ALD HfO_x are made for transistors from HfCl₄.

Same paragraph: ... vapor pressure is a crucial parameter for selecting precursors for ALD. Yes, volatility is the most important property. However, for practical use of precursors we need the knowledge at which temperature we can reach about 0.1 mbar pressure. Pressure is not a critical parameter in an ideal ALD process – the only thing is that we have enough precursor molecules to saturate the surface. Having said that it does not mean that knowing the vapor pressure behavior is not important but it can be more important outside ALD.

The experimental part is OK. The measurements are made carefully and the knowledge on isopropoxide is useful. It is also useful to see that the other alkoxides decompose.

Some formal comments:

Tables 5 and 9 are very short. Is it necessary to present the numbers in a table.

Numbers in Tables 6 and 7 are almost constant. Are the tables needed?

Text for Table 3 is very short and not understandable in present form.

Reviewer: 3

Comments to the Author(s)

This manuscript by Chen et al described the synthesis, characterization of two Hf alkoxides, and evaluated their potential in atomic layer deposition applications. The results and discussions are well organized. The manuscript is well written. However, due to the following reasons, major revision is recommended:

1. The authors claimed that they synthesized the Hf(OR)₄ compounds: 'the purity of hafnium alkoxides purified by reduced pressure distillation can be up to 99.997%.' It seems they did the calculation based on trace metal impurity. However, they didn't count organic impurities. If one checks the NMR spectra carefully (Figures 2a and 2b), there are obvious impure peaks that are adjacent to each product peak. This reviewer suspects there is unreacted alcohol. Did the authors do Elemental Analysis to compare with calculated C/H amounts? The authors likely used regular CDCl₃ as deuterated solvent instead of anhydrous CDCl₃. If the authors obtained high purity product, the impurity such as trace water may lead to decomposition and thus impurity peaks. The purity can't be so high as 99.997% if considering the impurities by NMR.
2. The authors mentioned previous work by Yang et al. on Hf tetrabutoxide. Please compare the results with Yang's and comment.
3. Figure 2a: the integrals are not correct and should be 4H and 24H.
4. Figure 2b: the integrals are not correct and should be 8H, 8H and 12H.

Author's Response to Decision Letter for (RSOS-180688.R0)

See Appendix B.

RSOS-181193.R0

Review form: Reviewer 3

Is the manuscript scientifically sound in its present form?

Yes

Are the interpretations and conclusions justified by the results?

No

Is the language acceptable?

Yes

Is it clear how to access all supporting data?

Yes

Do you have any ethical concerns with this paper?

No

Have you any concerns about statistical analyses in this paper?

No

Recommendation?

Accept with minor revision (please list in comments)

Comments to the Author(s)

Please see attached pdf copy of referee report (Appendix C).

Review form: Reviewer 4

Is the manuscript scientifically sound in its present form?

Yes

Are the interpretations and conclusions justified by the results?

No

Is the language acceptable?

Yes

Is it clear how to access all supporting data?

Not Applicable

Do you have any ethical concerns with this paper?

No

Have you any concerns about statistical analyses in this paper?

I do not feel qualified to assess the statistics

Recommendation?

Major revision is needed (please make suggestions in comments)

Comments to the Author(s)

In the work of Changhong Wang et al., in order to identify a volatile metallo-organic precursor for the deposition of HfO₂ films for atomic layer deposition (ALD) applications, the evaporative properties of three hafnium alkoxides were investigated using thermogravimetric analysis (TGA). Hafnium isopropoxide, hafnium n-propoxide and hafnium n-butoxide were synthesized by electrochemical method and characterized by Fourier transform infrared spectroscopy (FT-IR), nuclear magnetic resonance (NMR) and inductively coupled plasma (ICP) analysis techniques. Synthesized samples were subjected to a simultaneous thermogravimetric–differential thermal analysis (TG-DTA) unit at 10 K min⁻¹ in a dry nitrogen atmosphere flowing at 100 mL min⁻¹. And then, a modified Langmuir equation was used to calculate vapor pressure curves for hafnium isopropoxide and hafnium n-propoxide. The vapor pressure curve of hafnium isopropoxide was calculated to be $\ln p = 31.157(\pm 0.200) - 13130.57(\pm 56.50)/T$. However, no curve was constructed for hafnium n-propoxide and hafnium n-butoxide because these two compounds undergo evaporation and deposition simultaneously.

The manuscript is mostly well written, the method seems to be appropriate and properly conducted. However, explanations of the experimental procedures and results are not sufficient. Some more explanations need to be added to make the study more comprehensible. In addition,

the authors need some improvements to English, many sentences are absolutely not clear (see details below). The presentation of this work can be improved and have listed my suggestions below. At this stage, my recommendation for this manuscript is major revision.

Specific Comments:

1. Page 20, "Thermal property is of prime consideration.....predict the desired conditions and is favorable for process investigation." This sentence is too complicated and not clear.
2. Page 21, 3.2 characterization, the scan times and resolution of the FTIR measurements should be given.
3. Page 21, "Then, the distillation temperature was raised to corresponding temperature to remove a little amount of ester." What does the "corresponding temperature" mean? This sentence is not clear.
4. Page 21, "And then the impurity content of hafnium oxide was converted into that of hafnium alkoxides." I'm wondering where does the hafnium oxide comes from. More explanations should be added here.
5. In the whole text, both "°C" and "K" were used for temperature, which should be revised to maintain consistency.
6. Page 21, "the direct electrochemical synthesis of hafnium alkoxides compared with the traditional method has a greater promise owing to its remarkable advantages." "compared with the traditional method" can be deleted.
7. Page 21, "Detailed reasons were described previously in our study." "study" should be "studies".
8. Page 21, "hafnium alkoxides was successfully obtained by Turevskaya et al". "et al" should be "et al."
9. Page 21, TGA, FT-IR, 1H-NMR and ICP appeared in the main body for the first time without full name.
10. Page 22, "Peaks at around 2970 cm⁻¹ and 2860 cm⁻¹ correspond to asymmetric and symmetric stretch vibrations, respectively." Which functional group does these two peaks belong to?
11. Page 23, "Corresponding datum, which have been used to construct Figure 4, to calculate k are given in table 3, in which the values of pcalc. are obtained from the fitting line of plot of p against v." This sentence doesn't read smoothly.

Decision letter (RSOS-181193.R0)

18-Sep-2018

Dear Dr Wang:

Title: Determination of the vapor pressure curves and vaporization enthalpies of hafnium alkoxides using thermogravimetric analysis

Manuscript ID: RSOS-181193

The editor assigned to your paper has now received comments from reviewers. We would like you to revise your paper in accordance with the referee and Subject Editor suggestions which can be found below (not including confidential reports to the Editor). Please note this decision does not guarantee eventual acceptance.

Please submit a copy of your revised paper before 11-Oct-2018. Please note that the revision

deadline will expire at 00.00am on this date. If we do not hear from you within this time then it will be assumed that the paper has been withdrawn. In exceptional circumstances, extensions may be possible if agreed with the Editorial Office in advance. We do not allow multiple rounds of revision so we urge you to make every effort to fully address all of the comments at this stage. If deemed necessary by the Editors, your manuscript will be sent back to one or more of the original reviewers for assessment. If the original reviewers are not available we may invite new reviewers.

Yours sincerely,
Dr Laura Smith, MRSC
Publishing Editor, Journals
Royal Society of Chemistry,
Thomas Graham House,
Science Park, Milton Road,
Cambridge, CB4 0WF, UK

Royal Society Open Science - Chemistry Editorial Office

On behalf of the Subject Editor Professor Anthony Stace and the Associate Editor Professor Hazel Cox.

RSC Associate Editor
Comments to the Author:
(There are no comments.)

Reviewers' Comments to Author:
Reviewer: 3

Comments to the Author(s)
Please see attached pdf copy of referee report

Reviewer: 4

Comments to the Author(s)

In the work of Changhong Wang et al., in order to identify a volatile metallo-organic precursor for the deposition of HfO₂ films for atomic layer deposition (ALD) applications, the evaporative properties of three hafnium alkoxides were investigated using thermogravimetric analysis (TGA). Hafnium isopropoxide, hafnium n-propoxide and hafnium n-butoxide were synthesized by electrochemical method and characterized by Fourier transform infrared spectroscopy (FT-IR), nuclear magnetic resonance (NMR) and inductively coupled plasma (ICP) analysis techniques. Synthesized samples were subjected to a simultaneous thermogravimetric–differential thermal analysis (TG–DTA) unit at 10 K min⁻¹ in a dry nitrogen atmosphere flowing at 100 mL min⁻¹. And then, a modified Langmuir equation was used to calculate vapor pressure curves for hafnium isopropoxide and hafnium n-propoxide. The vapor pressure curve of hafnium isopropoxide was calculated to be $\ln p = 31.157(\pm 0.200) - 13130.57(\pm 56.50)/T$. However, no curve was constructed for hafnium n-propoxide and hafnium n-butoxide because these two compounds undergo evaporation and deposition simultaneously.

The manuscript is mostly well written, the method seems to be appropriate and properly conducted. However, explanations of the experimental procedures and results are not sufficient. Some more explanations need to be added to make the study more comprehensible. In addition, the authors need some improvements to English, many sentences are absolutely not clear (see details below). The presentation of this work can be improved and have listed my suggestions below. At this stage, my recommendation for this manuscript is major revision.

Specific Comments:

1. Page 20, "Thermal property is of prime consideration.....predict the desired conditions and is favorable for process investigation." This sentence is too complicated and not clear.
2. Page 21, 3.2 characterization, the scan times and resolution of the FTIR measurements should be given.
3. Page 21, "Then, the distillation temperature was raised to corresponding temperature to remove a little amount of ester." What does the "corresponding temperature" mean? This sentence is not clear.
4. Page 21, "And then the impurity content of hafnium oxide was converted into that of hafnium alkoxides." I'm wondering where does the hafnium oxide comes from. More explanations should be added here.
5. In the whole text, both "°C" and "K" were used for temperature, which should be revised to maintain consistency.
6. Page 21, "the direct electrochemical synthesis of hafnium alkoxides compared with the traditional method has a greater promise owing to its remarkable advantages." "compared with the traditional method" can be deleted.
7. Page 21, "Detailed reasons were described previously in our study." "study" should be "studies".
8. Page 21, "hafnium alkoxides was successfully obtained by Turevskaya et al". "et al" should be "et al."
9. Page 21, TGA, FT-IR, 1H-NMR and ICP appeared in the main body for the first time without full name.
10. Page 22, "Peaks at around 2970 cm⁻¹ and 2860 cm⁻¹ correspond to asymmetric and symmetric stretch vibrations, respectively." Which functional group does these two peaks belong to?
11. Page 23, "Corresponding datum, which have been used to construct Figure 4, to calculate k are given in table 3, in which the values of p_{calc.} are obtained from the fitting line of plot of p against v." This sentence doesn't read smoothly.

Author's Response to Decision Letter for (RSOS-181193.R0)

See Appendix D.

RSOS-181193.R1 (Revision)

Review form: Reviewer 3

Is the manuscript scientifically sound in its present form?

Yes

Are the interpretations and conclusions justified by the results?

Yes

Is the language acceptable?

Yes

Is it clear how to access all supporting data?

Yes

Do you have any ethical concerns with this paper?

No

Have you any concerns about statistical analyses in this paper?

I do not feel qualified to assess the statistics

Recommendation?

Accept with minor revision (please list in comments)

Comments to the Author(s)

The comments have been addressed except the following: In NMR analysis section, please add explanation of source of the impure peaks into the manuscript as the authors did in the rebuttal.

Review form: Reviewer 4

Is the manuscript scientifically sound in its present form?

Yes

Are the interpretations and conclusions justified by the results?

Yes

Is the language acceptable?

No

Is it clear how to access all supporting data?

No

Do you have any ethical concerns with this paper?

No

Have you any concerns about statistical analyses in this paper?

No

Recommendation?

Accept with minor revision (please list in comments)

Comments to the Author(s)

I have checked the revised manuscript carefully. Further discussion about the removal of impurities contents of hafnium alkoxides were added. In addition, more explanations were given which made the manuscript complete and comprehensible. However, it seems that the authors have only revised the manuscript according to part of the suggestions I have raised.

1. Although I have raised that some sentences in the manuscript are too complicated and not clear, the authors just did a few punctuation tweaks in the revised version.

2. Once again, the scan times of the FTIR measurements should be given for clear presentation. After the modification made by the authors, the revised manuscript is much better than the previous edition. However, the authors should carefully check the format and English of the article. It would be a qualified manuscript after the authors take care of these details.

Decision letter (RSOS-181193.R1)

13-Nov-2018

Dear Dr Wang:

Title: Determination of the vapor pressure curves and vaporization enthalpies of hafnium alkoxides using thermogravimetric analysis

Manuscript ID: RSOS-181193.R1

Thank you for submitting the above manuscript to Royal Society Open Science. On behalf of the Editors and the Royal Society of Chemistry, I am pleased to inform you that your manuscript will be accepted for publication in Royal Society Open Science subject to minor revision in accordance with the referee suggestions. Please find the reviewers' comments at the end of this email.

The reviewers and handling editors have recommended publication, but also suggest some minor revisions to your manuscript. Therefore, I invite you to respond to the comments and revise your manuscript.

Because the schedule for publication is very tight, it is a condition of publication that you submit the revised version of your manuscript before 22-Nov-2018. Please note that the revision deadline will expire at 00.00am on this date. If you do not think you will be able to meet this date please let me know immediately.

To revise your manuscript, log into <https://mc.manuscriptcentral.com/rsos> and enter your Author Centre, where you will find your manuscript title listed under "Manuscripts with

Decisions". Under "Actions," click on "Create a Revision." You will be unable to make your revisions on the originally submitted version of the manuscript. Instead, revise your manuscript and upload a new version through your Author Centre.

Best wishes,

Dr Laura Smith
Publishing Editor, Journals

On behalf of the Subject Editor Professor Anthony Stace and the Associate Editor Professor Hazel Cox.

RSC Associate Editor:
Comments to the Author:
(There are no comments.)

RSC Subject Editor:
Comments to the Author:
(There are no comments.)

Reviewer comments to Author:
Reviewer: 3

Comments to the Author(s)
The comments have been addressed except the following: In NMR analysis section, please add explanation of source of the impure peaks into the manuscript as the authors did in the rebuttal.

Reviewer: 4

Comments to the Author(s)
I have checked the revised manuscript carefully. Further discussion about the removal of impurities contents of hafnium alkoxides were added. In addition, more explanations were given which made the manuscript complete and comprehensible. However, it seems that the authors have only revised the manuscript according to part of the suggestions I have raised.

1. Although I have raised that some sentences in the manuscript are too complicated and not clear, the authors just did a few punctuation tweaks in the revised version.
2. Once again, the scan times of the FTIR measurements should be given for clear presentation. After the modification made by the authors, the revised manuscript is much better than the previous edition. However, the authors should carefully check the format and English of the article. It would be a qualified manuscript after the authors take care of these details.

Author's Response to Decision Letter for (RSOS-181193.R1)

See Appendix E.

Decision letter (RSOS-181193.R2)

03-Dec-2018

Dear Dr Wang:

Title: Determination of the vapor pressure curves and vaporization enthalpies of hafnium alkoxides using thermogravimetric analysis

Manuscript ID: RSOS-181193.R2

It is a pleasure to accept your manuscript in its current form for publication in Royal Society Open Science. The chemistry content of Royal Society Open Science is published in collaboration with the Royal Society of Chemistry.

On behalf of the Subject Editor Professor Anthony Stace and the Associate Editor Professor Hazel Cox.

RSC Associate Editor
Comments to the Author:
(There are no comments.)

Reviewer(s)' Comments to Author:

Appendix A

The vapor pressure and vaporization enthalpy for metal-organic compounds is an indispensable part of application of such systems in CVD or ALD processes. At the same time the incorrect data or incorrectly assessed values can bring too many troubles for elimination of the substances for CVD or ALD. Thus the experimental values determined in the current study of rather high importance but should be significantly improved in the field of assessment of the studied process and uncertainty of the final values. The following issues should be resolved before the article can be published.

1. Abstract: What are “the evaporative properties”?
2. Abstract: Synthesized samples were subject to a simultaneous TGA in a nitrogen atmosphere at $10\text{ }^{\circ}\text{C min}^{-1}$ at 100 mL min^{-1} .
This sentence is not clear. What do authors mean under simultaneous TGA? And does it mean “were subject to TGA”?
3. Introduction:
“In early years, TG is used in an isothermal condition, but recently an elevated temperature program is used, which makes the obtainment of data faster and reduces the number of samples required [7, 8]” Thermogravimetry from the very beginning was used in scanning regime at elevated temperatures? The idea of this sentence is absolutely not clear or poor.
4. Introduction: “this HfO_2 layers” change for “these HfO_2 layers”
5. Introduction: “Vapor pressure curves can be found using the Antoine equation.” Antoine equation is only the fitting equation for vapor pressure and can’t serve as the origin of vapor pressure data for compounds.
6. Table 1: please state the origin of provided data (certificate or method of determination)
7. 2.1. Materials and sample preparation:
“Electrochemical synthesis of the hafnium alkoxides was carried out under the basic experimental conditions: anhydrous alcohol 2.2 L, conductive agent Et_4NBr 0.04 M, required solution temperature (boiling temperature), polar distance 2 cm and applied current 2A (current density 100 A m^{-2}).”
Why authors call that the basic conditions? At what temperature the synthesis was carried out? Here and in further synthesis and distillation methodology description state explicitly the temperature of the process.
8. 2.2. Characterization:
“After desiccation in desiccator at $100\text{ }^{\circ}\text{C}$ for 12 h.”
What was used for sample drying? Vacuum conditions or chemical reagent? In any case the conditions should be obviously presented.
9. 2.3. Thermogravimetric analysis
“Vapor pressure curves can be obtained using the Antoine equation [18]:”
The same as point 5.
10. 2.3. Thermogravimetric analysis:
“The principle of using thermogravimetry to estimate the vapor pressure is based on the Langmuir equation [19]:”
Irvine Langmuir has proposed this equation for vacuum conditions for the case of molecular flow rate from the sample surface. Please indicate why this equation was used for atmospheric pressures.
11. 2.3. Thermogravimetric analysis:
Langmuir equation. Why condensation coefficient is mentioned in left and right side of equation?

12. 2.3. Thermogravimetric analysis:

“Benzoic acid has been assumed as a suitable material for this role [20, 21]. The enthalpy of vaporization is calculated from the vapor pressure–temperature data using the Clausius-Clapeyron equation:”

The use of reference material is widely used approach. But the vapor pressure data and vaporization enthalpy values should be explicitly presented in the text of the article, as well as, the experimentally determined vaporization enthalpy for this compound.

13. 3.2.1 Hafnium isopropoxide:

“Using the p-T data, plot of $\ln p$ versus $1/T$ is obtained as straight line for hafnium isopropoxide, from which, using the Eq. (4), the slope gives $-\Delta H_{\text{vap}}/R$, and hence the vaporization enthalpy (ΔH_{vap}). The plot of $\ln p$ versus $1/T$ for hafnium isopropoxide and its linear fitting are shown in Fig. 5, and from the slope the enthalpy of evaporation is calculated to be $110.5 \pm 0.39 \text{ kJ mol}^{-1}$ ”

1. Apparently, the experimental vapor pressure dependence depicted in Fig 5 is not linear but convex. In this case the additional heat $\Delta C_{p,\text{vap}}$ should be applied or the effect of the vaporization surface decrease with the mass loss should be accounted.

2. The vapor pressure calculated according to the Langmuir equation has the molar mass of vapors as one of the components. In the case of salts like studied hafnium alkoxides the gas phase consists of the wide variety of associated particles like monomeric form, dimers, trimers and so on. The vaporization of each should be treated separately while the composition of the gas phase will change with temperature. The best way is to carry out in situ analysis of the gas phase (for example with Mass spectrometry with low energy of electron impact) in order to assess the vaporization enthalpy and vapor pressure correctly. The heating rate and gas flow rate will not affect the association composition. And thus, carried out investigation does not clarify the gas phase state of hafnium alkoxides, what is of crucial importance for ALD process.

14. 3.2 Thermal properties analysis and vapor pressure estimation:

For all compounds under study the simultaneous TGA/DTG/DTA was applied. The presence of two or more peaks was used for assessment of the process under study (vaporization or decomposition). At the same time the same two peaks were observed for benzoic acid in DTA. Does it mean that the sample of benzoic acid also decomposed at rather mild conditions like 200 C?

Vaporization and decomposition are not the same as melting. They take place at each temperature and if it obviously presented 360 C it does not mean that it not observed below this temperature. It can have the lower rate the vaporization but still have a big part in the final mass loss rate. Therefore, all investigations of this type should be combined with physical or chemical analysis of the gas phase in the temperature range of investigation.

15. For all experimental values the type of uncertainty (standard or extended) and corresponding level of confidence should be given.

16. It is better to find the English native speaker in order to resolve possible issues with English throughout the text. Many sentences are too complicated and not clear.

Appendix B

Manuscript ID: RSOS-180688

Title: Determination of the vapor pressure curves and vaporization enthalpies of hafnium alkoxides using thermogravimetric analysis

RSOS

REVIEWER(S) REPORTS:

Associate Editor Comments to Author ():

RSC Associate Editor:

Comments to the Author:

All of the reviewers comments need to be considered and addressed in detail before it can be reconsidered for publication.

RSC Subject Editor:

Comments to the Author:

(There are no comments.)

To editor:

Dear Editor,

Thank you very much for sending us the ID RSOS-180688 Decision Letter on 18-Jun-2018. We really appreciate your advice and the reviewer's comments to our manuscript. We have studied all comments carefully, and thank you very much for your time and thoughtful comments. We have now revised the manuscript in accordance with the comments and suggestions. **All changes made to the text are highlighted with a red color.** The entire comments are responded point by point as follows.

To reviewer 1:

Thank you very much for your time and thoughtful comments again. If I remember correctly, I've received a same review. Before this paper was submitted to RSOS, we had studied all comments reviewer 1 put forward carefully and had revised the manuscript in accordance with the constructive comments and suggestions.

Comments:

The vapor pressure and vaporization enthalpy for metal-organic compounds is an indispensable part of application of such systems in CVD or ALD processes. At the same time the incorrect data or incorrectly assessed values can bring too many

troubles for elimination of the substances for CVD or ALD. Thus the experimental values determined in the current study of rather high importance but should be significantly improved in the field of assessment of the studied process and uncertainty of the final values. The following issues should be resolved before the article can be published.

Comment 1: Abstract: What are “the evaporative properties”?

Answer 1: The evaporative properties refer to the properties of converting from liquid to gas. Their evaluation indexes include boiling range and vapor pressure.

Comment 2: Abstract: Synthesized samples were subject to a simultaneous TGA in a nitrogen atmosphere at $10\text{ }^{\circ}\text{C min}^{-1}$ at 100 mL min^{-1} .

This sentence is not clear. What do authors mean under simultaneous TGA? And does it mean

“were subject to TGA”?

Answer 2: Thank you for pointing this out. Corresponding sentence had been revised to “Synthesized samples were subject to a simultaneous thermogravimetric–differential thermal analysis (TG–DTA) at 10 K min^{-1} in a dry nitrogen atmosphere flowing at 100 mL min^{-1} ”.

Comment 3: Introduction: “In early years, TG is used in an isothermal condition, but recently an elevated temperature program is used, which makes the obtainment of data faster and reduces the number of samples required [7, 8]” Thermogravimetry from the very beginning was used in scanning regime at elevated temperatures? The idea of this sentence is absolutely not clear or poor.

Answer 3: According to reviewer’s comment, corresponding sentence had been revised to “In earlier years, TG was used in an isothermal condition, but more recently, an increasing temperature heating program, which makes the obtainment of data faster and reduces the number of samples required [7,8], has been used in TG. ”

Comment 4: Introduction: “this HfO₂ layers” change for “these HfO₂ layers”.

Answer 4: According to reviewer’s comment, corresponding sentence had been revised.

Comment 5: Introduction: “Vapor pressure curves can be found using the Antoine equation.” Antoine equation is only the fitting equation for vapor pressure and can’t serve as the origin of vapor pressure data for compounds.

Answer 5: Thank you very much for valuable suggestion. Corresponding sentence had been revised.

Comment 6: Table 1: please state the origin of provided data (certificate or method of determination)

Answer 6: According to reviewer’s constructive comment, method of determination had been added to manuscript: the hafnium plates were made from hafnium powders of metallurgical grade with self-resistance sintering, electron bombardment and rolling treatment. In order to determine its chemical composition, hafnium plate was dissolved in hydrofluoric acid of a required concentration. The impurity contents were measured by ICP-Mass Agilent 7500a analyzer, and the corresponding result was presented in Table 1 of manuscript.

Comment 7: 2.1. Materials and sample preparation:

“Electrochemical synthesis of the hafnium alkoxides was carried out under the basic experimental conditions: anhydrous alcohol 2.2 L, conductive agent Et₄NBr 0.04 M, required solution temperature (boiling temperature), polar distance 2 cm and applied current 2A (current density 100A m⁻²).”

Why authors call that the basic conditions? At what temperature the synthesis was carried out? Here and in further synthesis and distillation methodology description state explicitly the temperature of the process.

Answer 7: Thank you for pointing this out. To describe it accurately, I had changed “the basic conditions” to “the fixed condition”. The reason why we call that the basic conditions before was that the condition under which hafnium alkoxides was synthesized in the paper was determined based on the process-optimization experiments we had carried out. The temperatures of the synthesis had been added to the paper: The synthesis of hafnium isopropoxide, hafnium *n*-propoxide and hafnium *n*-butoxide was carried out at 82.4 °C, 97.1 °C and 117.7 °C, respectively.

Comment 8: 2.2. Characterization:

“After desiccation in desiccator at 100 °C for 12 h.”

What was used for sample drying? Vacuum conditions or chemical reagent? In any case the conditions should be obviously presented.

Answer 8: According to the reviewer's constructive instruction, corresponding part had been revised. Anhydrous magnesium sulfate was used for sample drying. Desiccation in desiccator was at 100 °C for 12 h under vacuum condition.

Comment 9: 2.3. Thermogravimetric analysis

“Vapor pressure curves can be obtained using the Antoine equation [18]:”

The same as point 5.

Answer 9: According to the reviewer's suggestion, corresponding sentence had been revised.

Comment 10: 2.3. Thermogravimetric analysis:

“The principle of using thermogravimetry to estimate the vapor pressure is based on the Langmuir equation [19]:”

Irvine Langmuir has proposed this equation for vacuum conditions for the case of molecular flow rate from the sample surface. Please indicate why this equation was used for atmospheric pressures.

Answer 10: Thank you for pointing this out. Actually, estimating the vapor pressure using thermogravimetry based on the Langmuir equation was carried out on the basis of related literatures [1-3 etc.]. I am sorry to have no idea about the reason. Please enlighten us if possible.

[1] Wright S F, Dollimore D, Dunn J G, et al. Determination of the vapor pressure curves of adipic acid and triethanolamine using thermogravimetric analysis. *Thermochimica Acta*, 2004, 421(1):25-30.

[2] Wright S F, Phang P, Dollimore D, et al. An overview of calibration materials used in thermal analysis—benzoic acid. *Thermochimica Acta*, 2002, 392(02):251-257.

[3] Gomes A P B, Freire F D, Aragão C F S. Determination of vapor pressure curves of warifteine and methylwarifteine by using thermogravimetry. *Journal of Thermal Analysis & Calorimetry*, 2012, 108(1):249-252.

Comment 11: 2.3. Thermogravimetric analysis:

Langmuir equation. Why condensation coefficient is mentioned in left and right side of equation?

Answer 11: Thank you for pointing this out. a in left side of the equation is unit area (m^2), while α in right side of the equation is condensation coefficient.

Comment 12: 2.3. Thermogravimetric analysis:

“Benzoic acid has been assumed as a suitable material for this role [20, 21]. The enthalpy of vaporization is calculated from the vapor pressure–temperature data using the Clausius-Clapeyron equation:”

The use of reference material is widely used approach. But the vapor pressure data and vaporization enthalpy values should be explicitly presented in the text of the article, as well as, the experimentally determined vaporization enthalpy for this compound.

Answer 12: According to reviewer's constructive instruction, corresponding part had been revised. The effect of temperature on the vapor pressure of benzoic acid can be expressed by Eq. (1):

$$\log p = A - B/(T + C)$$

(1)

Where A–C in the temperature range of 405 to 523 K are 7.80991, 2776.12, and 43.978, respectively [4].

[4] Wright S F, Phang P, Dollimore D, et al. An overview of calibration materials used in thermal analysis—benzoic acid. *Thermochimica Acta*, 2002, 392(02):251-257.

Comment 13: 3.2.1 Hafnium isopropoxide:

“Using the p-T data, plot of $\ln p$ versus $1/T$ is obtained as straight line for hafnium isopropoxide, from which, using the Eq. (4), the slope gives $-\Delta H_{\text{vap}}/R$, and hence the vaporization enthalpy (ΔH_{vap}). The plot of $\ln p$ versus $1/T$ for hafnium isopropoxide and its linear fitting are shown in Fig. 5, and from the slope the enthalpy of evaporation is calculated to be $110.5 \pm 0.39 \text{ kJ mol}^{-1}$ ”

1. Apparently, the experimental vapor pressure dependence depicted in Fig 5 is not linear but convex. In this case the additional heat $C_{p_{\text{vap}}}$ should be applied or the effect of the vaporization surface decrease with the mass loss should be accounted.

2. The vapor pressure calculated according to the Langmuir equation has the molar mass of vapors as one of the components. In the case of salts like studied hafnium alkoxides the gas phase consists of the wide variety of associated particles like monomeric form, dimers, trimmers and so on. The vaporization of each should be

treated separately while the composition of the gas phase will change with temperature. The best way is to carry out in situ analysis of the gas phase (for example with Mass spectrometry with low energy of electron impact) in order to assess the vaporization enthalpy and vapor pressure correctly. The heating rate and gas flow rate will not affect the association composition. And thus, carried out investigation does not clarify the gas phase state of hafnium alkoxides, what is of crucial importance for ALD process.

Answer 13: According to reviewer's valuable instruction, corresponding part had been revised. In order to assess the vaporization enthalpy and vapor pressure correctly, mass spectra of the gas phase were obtained on MSQ8100 spectrometers (direct introduction, 70 eV) at required temperature range. The mass-spectra are given in Table 1, which indicates the presence of trimeric molecules in the gas phase.

Table 1 Mass-spectra of “Hf(OR)₄”

“Hf(OPr ⁱ) ₄ ”	“Hf(OPr ⁿ) ₄ ”	“Hf(OBu ⁿ) ₄ ”	interpretation
m/z			
1240			[Hf ₃ (OR) ₁₂] ⁺
1180	1181	1332	[Hf ₃ (OR) ₁₁] ⁺
	1171		[Hf ₃ (OR) ₁₀ (OEt)] ⁺
		1295	[Hf ₃ (OR) ₁₀ (OCH ₂)] ⁺
1143	1140	1274	[Hf ₃ O(OR) ₁₀] ⁺
1085	1088	1210	[Hf ₃ O(OR) ₉] ⁺
	1111		[Hf ₃ O(OR) ₉ (OCH ₂)] ⁺
	1073		[Hf ₃ O(OR) ₈ (OEt)] ⁺
		1160	[Hf ₃ O(OR) ₈ (OMe)] ⁺
980	980	1071	[Hf ₃ O ₂ (OR) ₇] ⁺
		1068	[Hf ₃ O ₂ (OR) ₆ (OPr)] ⁺
	976	1047	[Hf ₃ O ₂ (OR) ₆ (OEt)] ⁺
879	876	942	[Hf ₃ O ₃ (OR) ₅] ⁺
	814		[Hf ₃ O ₃ (OR) ₄] ⁺
763		807	[Hf ₃ O ₃ (OR) ₃] ⁺
719	721		[Hf ₃ O ₄ (OR) ₂] ⁺
763	758	854	[Hf ₂ (OR) ₇] ⁺
		679	[Hf ₂ O(OR) ₄ (OH)] ⁺
	667	737	[Hf ₂ O(OR) ₅] ⁺

567	569	609	[Hf ₂ O ₂ (OR) ₃] ⁺
	404	461	[Hf(OR) ₄] ⁺
		453	[Hf(OR) ₃ (OC ₃ H ₆)] ⁺
		425	[Hf(OR) ₃ (OCH ₂)] ⁺
358		400	[Hf(OR) ₃] ⁺
		369	[Hf(OR) ₂ (OEt)] ⁺
		324	[Hf(OR) ₂] ⁺

Comment 14: 3.2 Thermal properties analysis and vapor pressure estimation:

For all compounds under study the simultaneous TGA/DTG/DTA was applied. The presence of two or more peaks was used for assessment of the process under study (vaporization or decomposition). At the same time the same two peaks were observe for benzoic acid in DTA. Does it mean that the sample of benzoic acid also decomposed at rather mild conditions like 200 C?

Vaporization and decomposition are not the same as melting. They take place at each temperature and if it obviously presented 360 C it does not mean that it not observed below this temperature. It can have the lower rate the vaporization but still have a big part in the final mass loss rate. Therefore, all investigations of this type should be combined with physical or chemical analysis of the gas phase in the temperature range of investigation.

Answer 14: According to reviewer's constructive instruction, corresponding part had been revised. In the DTA plot for benzoic acid, there was one endothermic peak in a lower temperature region, which is taken to be the melting point (T_m) and one endothermic peak in the upper temperature region, which is taken to represent evaporation [4]. In order to determine whether the decomposition of the "Hf(OR)₄" below the temperature mentioned above occurred, mass-spectra of "Hf(OR)₄" at given evaporation temperature ranges had been carried out. Combined with the TG findings that for hafnium isopropoxide, trace white residues appeared on the TG crucible at the ending of evaporation range, while for hafnium *n*-propoxide and hafnium *n*-butoxide, a major amount of white residues appeared on the TG crucible at the ending of evaporation range, it can be deduced that the decomposition was likely to be much slower at the lower temperatures for hafnium isopropoxide, and that hafnium *n*-propoxide and hafnium *n*-butoxide were simultaneously undergoing evaporation and significant decomposition at evaporation range.

[4] Wright S F, Phang P, Dollimore D, et al. An overview of calibration materials used in thermal analysis—benzoic acid. *Thermochimica Acta*, 2002, 392(02):251-

257.

Comment 15: For all experimental values the type of uncertainty (standard or extended) and corresponding level of confidence should be given.

Answer 15: According to reviewer's constructive instruction, corresponding part had been revised.

Comment 16: It is better to find the English native speaker in order to resolve possible issues with English throughout the text. Many sentences are too complicated and not clear.

Answer 16: According to the reviewer's constructive instruction, we had checked the manuscript carefully and tried our best to improve the expression of the whole manuscript. The whole manuscript had been revised by a professional English teacher as well.

To reviewer 2:

Comment 1: Introduction 2nd paragraph: Hafnium alkoxide is mainly used in ALD [10]. This not true and the ref. 10 does not say that. Ref. 10 tells about the density of the films. A better reference what precursors have been used in ALD (and also in case of Hf) is J. Appl. Phys. 113 (2013) 021301.

Answer 1: According to the reviewer's constructive suggestion, corresponding reference has been changed in the revised manuscript.

Comment 2: The reference 11 is also bad if a general reference is wanted to show the importance of HfO₂ as high-*k* material. There are more than 5 000 papers on this topic in the literature and best is to refer to a review. A good example could be Mater. Sci. Reports 72 (2011) 97. Just to your information, in industrial scale ALD HfO_x are made for transistors from HfCl₄.

Answer 2: According to the reviewer's valuable suggestion, corresponding reference has been changed in the revised manuscript.

Comment 3: Same paragraph: ... vapor pressure is a crucial parameter for selecting precursors for ALD. Yes, volatility is the most important property. However, for practical use of precursors we need the knowledge at which

temperature we can reach about 0.1 mbar pressure. Pressure is not a critical parameter in an ideal ALD process – the only thing is that we have enough precursor molecules to saturate the surface. Having said that it does not mean that knowing the vapor pressure behavior is not important but it can be more important outside ALD.

Answer 3: Thank you for pointing this out.

Comment 4: The experimental part is OK. The measurements are made carefully and the knowledge on isopropoxide is useful. It is also useful to see that the other alkoxides decompose.

Answer 4: Thank you for pointing this out.

Comment 5: Tables 5 and 9 are very short. Is it necessary to present the numbers in a table. Numbers in Tables 6 and 7 are almost constant. Are the tables needed? Text for Table 3 is very short and not understandable in present form.

Answer 5: According to reviewer's suggestion, corresponding part has been revised. Tables 5 and 9 show the parameters of the fitting line ($y=a+bx$) for hafnium isopropoxide and hafnium *n*-propoxide in the plot of $\ln p$ against $1/T$, respectively. The datum in Table 5 and 9 correspond to those in Figure 6 and 9, respectively. Table 6 presents the effect of different heating rates on the vaporization enthalpies of hafnium isopropoxide at a constant nitrogen flow rate of 100 mL min^{-1} . Table 7 presents the effect of different nitrogen flow rates on the vaporization enthalpies of hafnium isopropoxide at a constant heating rate of 10 K min^{-1} . These Tables were made to search through datum easily. Table 3 records the corresponding datum to calculate k using eq. 1. The datum in Table 3 correspond to those in Figure 4.

$$p=kv \tag{1}$$

where $k = \sqrt{2\pi R} / \alpha$ and $v=(1/a)(dm/dt)\sqrt{T/M}$.

To reviewer 3:

Comments to the Author(s)

This manuscript by Chen et al described the synthesis, characterization of two Hf alkoxides, and evaluated their potential in atomic layer deposition applications. The results and discussions are well organized. The manuscript is well written. However, due to the following reasons, major revision is recommended:

Comment 1: The authors claimed that they synthesized the Hf(OR)₄ compounds: ‘the purity of hafnium alkoxides purified by reduced pressure distillation can be up to 99.997%.’ It seems they did the calculation based on trace metal impurity. However, they didn’t count organic impurities. If one checks the NMR spectra carefully (Figures 2a and 2b), there are obvious impure peaks that are adjacent to each product peak. This reviewer suspects there is unreacted alcohol. Did the authors do Elemental Analysis to compare with calculated C/H amounts? The authors likely used regular CDCl₃ as deuterated solvent instead of anhydrous CDCl₃. If the authors obtained high purity product, the impurity such as trace water may lead to decomposition and thus impurity peaks. The purity can’t be so high as 99.997% if considering the impurities by NMR.

Answer 1: Thank you for pointing this out. The Hf(OR)₄ are acutely sensitive to water and easy to suffer hydrolysis. The process can be expressed as follows:

So it’s hard to determine its contents by ICP or other methods measuring the sample in aqueous solution. Therefore, in order to determine impurity contents of Hf(OR)₄, some amount of water was added into Hf(OR)₄ solution for hydrolysis reaction. After desiccation in desiccator under vacuum conditions at 100 °C for 12 h, the sample was calcined in muffle furnace at 800 °C for 2 h. The impurity contents were obtained by ICP-Mass Agilent 7500a analyzer. And then the impurity content of hafnium oxide was converted into that of Hf(OR)₄. In addition, mass spectra of Hf(OR)₄ were obtained on MSQ8100 spectrometers (direct introduction, 70 eV) at required temperature range. The mass-spectra are given in Table 1, which indicates the presence of pure trimeric molecules. Furthermore, No bands can be observed within the range 3600-3100 cm⁻¹ in Figure 1 of manuscript. This indicates that the samples were well preserved and the partial hydrolyzation has not occurred, and, thus, there should be no unreacted alcohol. As for CDCl₃ mentioned above, I have no idea about it. Please enlighten us if possible.

Table 1 Mass-spectra of “Hf(OR)₄”

“Hf(OPr ^{i}) ₄ ”	“Hf(OPr ^{n}) ₄ ”	“Hf(OBu ^{n}) ₄ ”	interpretation
m/z			
1240			[Hf ₃ (OR) ₁₂] ⁺
1180	1181	1332	[Hf ₃ (OR) ₁₁] ⁺
	1171		[Hf ₃ (OR) ₁₀ (OEt)] ⁺

		1295	$[\text{Hf}_3(\text{OR})_{10}(\text{OCH}_2)]^+$
1143	1140	1274	$[\text{Hf}_3\text{O}(\text{OR})_{10}]^+$
1085	1088	1210	$[\text{Hf}_3\text{O}(\text{OR})_9]^+$
	1111		$[\text{Hf}_3\text{O}(\text{OR})_9(\text{OCH}_2)]^+$
	1073		$[\text{Hf}_3\text{O}(\text{OR})_8(\text{OEt})]^+$
		1160	$[\text{Hf}_3\text{O}(\text{OR})_8(\text{OMe})]^+$
980	980	1071	$[\text{Hf}_3\text{O}_2(\text{OR})_7]^+$
		1068	$[\text{Hf}_3\text{O}_2(\text{OR})_6(\text{OPr})]^+$
	976	1047	$[\text{Hf}_3\text{O}_2(\text{OR})_6(\text{OEt})]^+$
879	876	942	$[\text{Hf}_3\text{O}_3(\text{OR})_5]^+$
	814		$[\text{Hf}_3\text{O}_3(\text{OR})_4]^+$
763		807	$[\text{Hf}_3\text{O}_3(\text{OR})_3]^+$
719	721		$[\text{Hf}_3\text{O}_4(\text{OR})_2]^+$
763	758	854	$[\text{Hf}_2(\text{OR})_7]^+$
		679	$[\text{Hf}_2\text{O}(\text{OR})_4(\text{OH})]^+$
	667	737	$[\text{Hf}_2\text{O}(\text{OR})_5]^+$
567	569	609	$[\text{Hf}_2\text{O}_2(\text{OR})_3]^+$
	404	461	$[\text{Hf}(\text{OR})_4]^+$
		453	$[\text{Hf}(\text{OR})_3(\text{OC}_3\text{H}_6)]^+$
		425	$[\text{Hf}(\text{OR})_3(\text{OCH}_2)]^+$
358		400	$[\text{Hf}(\text{OR})_3]^+$
		369	$[\text{Hf}(\text{OR})_2(\text{OEt})]^+$
		324	$[\text{Hf}(\text{OR})_2]^+$

Comment 2: The authors mentioned previous work by Yang et al. on Hf tetrabutoxide. Please compare the results with Yang's and comment.

Answer 2: According to review's constructive suggestion, corresponding changes have been made in the revised manuscript. In Yang et al's paper, the deposition of hafnium oxide films on silicon substrates by MOCVD was investigated using the newly employed novel single precursor hafnium 3-methyl-3-pentoxide ($\text{Hf}(\text{mp})_4$). The reaction mechanism was clearly identified by the analysis of its thermal decomposition. The deposition characteristics were investigated together with the composition, crystallinity, and electrical properties of the deposited films. The properties of the $\text{Hf}(\text{mp})_4$ single precursor were compared with other hafnium alkoxide precursors. Probably because the authors mainly focus on the thermal property of $\text{Hf}(\text{mp})_4$, the thermal property of the hafnium *t*-butoxide is only simply

mentioned as a comparison. The result and discussion for hafnium *t*-butoxide quoted from the reference only has a sentence: “The hafnium *t*-butoxide was measured by TGA and was detected giving the residue of almost 30%, which is much more than that of Hf(mp)₄”. Compared to our results, the residue of hafnium *t*-butoxide (almost 30%) is higher than that of hafnium isopropoxide (almost 22%), but less than those of hafnium *n*-propoxide (44 %) and hafnium *n*-butoxide (41.6%), respectively.

Comment 3: Figure 2a: the integrals are not correct and should be 4H and 24H.

Comment 4: Figure 2b: the integrals are not correct and should be 8H, 8H and 12H.

Answer 3 and 4: Thank you for pointing this out. Figure 2 was plotted by MestReNova and the integral were calculated using this program.

Thanks to the referees for the thoughtful and thorough review. We have made careful modification on the original manuscript. We sincerely hope the present manuscript with the above revisions and replies is acceptable for publication in your journal.

Great thanks to you and the referee for the time and effort expended on this paper.

Best Wishes,

Sincerely yours,

Yongming Chen

Institute: Central South University

Address: No.932, Lushan Road, Changsha city, Hunan Province, China 410083

Tel: +86-18684685548

E-mail: csuchenyongming@163.com

Appendix C

It seems the authors are not familiar with NMR and MNova software, or this reviewer didn't make it clear enough. To use NMR in characterizing compounds with protons, one uses absolute proton numbers instead of the ratios as the authors insist using here, the latter just not acceptable to an organic/organometallic chemist.

Please click 'manual' integration, then put the cursor on peak A, right click and choose 'edit integral', then you may want to change 'normalized value' to the absolute proton numbers of peak A. Now, all the numbers you see will be the real (absolute) proton numbers.

Speaking of CDCl₃, the authors responded 'As for CDCl₃ mentioned above, I have no idea about it. Please enlighten us if possible.' First, that is the deuterated solvent the authors indicated in the following spectra, and 'chloroform-d' is a typo and should be 'chloroform-d'; second, usually CDCl₃ solvent is acidic and has trace amount of water, which sometimes causes decomposition or line broadening of the sample.

Hf(OR)₄ is far less sensitive than many organometallic compounds that my lab and all other organometallic chemistry labs work on. So it is absolutely an NMR that can be done relatively easily without hydrolysis or decomposition. Thus, this reviewer is not convinced whether the impurity NMR peaks come from unreacted starting materials or hydrolysis by impure deuterated solvent. No matter what, the NMR spectra show a mixture of at least two compounds for all the three cases.

Figure 2. ¹H NMR spectrum of hafnium alkoxide samples: (a) hafnium isopropoxide; (b) hafnium n-propoxide; (c) hafnium n-butoxide

287x201mm (96 x 96 DPI)

For this molecule, there are 4 protons for peak A and 24 protons for peak B. Please don't use the ratios! Why don't the authors change the integrals as I suggested and a chemist would do?

Figure 2. ^1H NMR spectrum of hafnium alkoxide samples: (a) hafnium isopropoxide; (b) hafnium n-propoxide; (c) hafnium n-butoxide

287x193mm (96 x 96 DPI)

Same problem. For this molecule, there are 8 protons for peak A, 8 protons for peak B and 12 protons for peak C.

Figure 2. ^1H NMR spectrum of hafnium alkoxide samples: (a) hafnium isopropoxide; (b) hafnium n-propoxide; (c) hafnium n-butoxide

287x191mm (96 x 96 DPI)

Same problem. For this molecule, there are 8 protons for peak A, 8 protons for peak B, 8 protons for peak C, and 12 protons for peak D.

Appendix D

Title: Determination of the vapor pressure curves and vaporization enthalpies of hafnium alkoxides using thermogravimetric analysis

Manuscript ID: RSOS-181193

To editor:

Dear Editor,

Thank you very much for sending us the ID RSOS-181193 Decision Letter on 18-Sep-2018. We really appreciate your advice and the reviewer's comments to our manuscript. We have studied all the comments carefully, and thank you very much for your time and thoughtful comments. We have now revised the manuscript in accordance with the comments and suggestions. **All changes made to the text are highlighted with a red color.** The entire comments are responded point by point as follows.

To reviewer 1:

Comments:

It seems the authors are not familiar with NMR and MNova software, or this reviewer didn't make it clear enough. To use NMR in characterizing compounds with protons, one uses absolute proton numbers instead of the ratios as the authors insist using here, the latter just not acceptable to an organic/organometallic chemist. Please click 'manual' integration, then put the cursor on peak A, right click and choose 'edit integral', then you may want to change 'normalized value' to the absolute proton numbers of peak A. Now, all the numbers you see will be the real (absolute) proton numbers.

Speaking of CDCl_3 , the authors responded 'As for CDCl_3 mentioned above, I have no idea about it. Please enlighten us if possible.' First, that is the deuterated solvent the authors indicated in the following spectra, and 'chloroform-d' is a typo and should be 'chloroform-d'; second, usually CDCl_3 solvent is acidic and has trace amount of water, which sometimes causes decomposition or line broadening of the sample.

$\text{Hf}(\text{OR})_4$ is far less sensitive than many organometallic compounds that my lab and all other organometallic chemistry labs work on. So it is absolutely an NMR that can be done relatively easily without hydrolysis or decomposition. Thus, this reviewer is not convinced whether the impurity NMR peaks come from unreacted starting materials or hydrolysis by impure deuterated solvent. No matter what, the NMR spectra show a mixture of at least two compounds for all the three cases.

Figure 2. ^1H NMR spectrum of hafnium alkoxide samples: (a) hafnium isopropoxide; (b) hafnium *n*-propoxide; (c) hafnium *n*-butoxide

For this molecule, there are 4 protons for peak A and 24 protons for peak B. Please don't use the ratios! Why don't the authors change the integrals as I suggested and a chemist would do?

Figure 2. ^1H NMR spectrum of hafnium alkoxide samples: (a) hafnium isopropoxide; (b) hafnium *n*-propoxide; (c) hafnium *n*-butoxide

Same problem. For this molecule, there are 8 protons for peak A, 8 protons for peak B and 12

protons for peak C.

Figure 2. ^1H NMR spectrum of hafnium alkoxide samples: (a) hafnium isopropoxide; (b) hafnium *n*-propoxide; (c) hafnium *n*-butoxide

Same problem. For this molecule, there are 8 protons for peak A, 8 protons for peak B, 8 protons for peak C, and 12 protons for peak D.

Answers: According to reviewer's constructive comment and instruction, corresponding parts have been revised. All of the integral values for peaks in ^1H NMR spectra have been changed to the absolute proton numbers using 'normalized value' function in the MNova software. All the typos of 'chloroform-d' in the figures have been revised to 'chloroform-d'. Details are presented in Figure 1. It is worth mentioning that mass spectra of $\text{Hf}(\text{OR})_4$ were obtained on MSQ8100 spectrometers (direct introduction, 70 eV) at required temperature range. The mass-spectra are given in Table 1, which indicates the presence of pure trimeric molecules. Furthermore, No bands can be observed within the range 3600-3100 cm^{-1} in Figure 1 of manuscript. This indicates that the samples were well preserved and the partial hydrolyzation has not occurred, and, thus, there should be no unreacted alcohol. Therefore, the impurity peak in the spectra may be attributed to the hydrolysis of $\text{Hf}(\text{OR})_4$ by the trace amount of water in CDCl_3 solvent since the $\text{Hf}(\text{OR})_4$ is acutely sensitive to water and easy to suffer hydrolysis.

Figure 1. ^1H NMR spectrum of hafnium alkoxide samples: (a) hafnium isopropoxide; (b) hafnium *n*-propoxide; (c) hafnium *n*-butoxide

Table 1 Mass-spectra of “ $\text{Hf}(\text{OR})_4$ ”

“ $\text{Hf}(\text{OPr}^i)_4$ ”	“ $\text{Hf}(\text{OPr}^n)_4$ ”	“ $\text{Hf}(\text{OBu}^n)_4$ ”	interpretation
m/z			
1240			$[\text{Hf}_3(\text{OR})_{12}]^+$
1180	1181	1332	$[\text{Hf}_3(\text{OR})_{11}]^+$
	1171		$[\text{Hf}_3(\text{OR})_{10}(\text{OEt})]^+$
		1295	$[\text{Hf}_3(\text{OR})_{10}(\text{OCH}_2)]^+$
1143	1140	1274	$[\text{Hf}_3\text{O}(\text{OR})_{10}]^+$
1085	1088	1210	$[\text{Hf}_3\text{O}(\text{OR})_9]^+$
	1111		$[\text{Hf}_3\text{O}(\text{OR})_9(\text{OCH}_2)]^+$
	1073		$[\text{Hf}_3\text{O}(\text{OR})_8(\text{OEt})]^+$
		1160	$[\text{Hf}_3\text{O}(\text{OR})_8(\text{OMe})]^+$
980	980	1071	$[\text{Hf}_3\text{O}_2(\text{OR})_7]^+$
		1068	$[\text{Hf}_3\text{O}_2(\text{OR})_6(\text{OPr})]^+$
	976	1047	$[\text{Hf}_3\text{O}_2(\text{OR})_6(\text{OEt})]^+$
879	876	942	$[\text{Hf}_3\text{O}_3(\text{OR})_5]^+$
	814		$[\text{Hf}_3\text{O}_3(\text{OR})_4]^+$
763		807	$[\text{Hf}_3\text{O}_3(\text{OR})_3]^+$
719	721		$[\text{Hf}_3\text{O}_4(\text{OR})_2]^+$
763	758	854	$[\text{Hf}_2(\text{OR})_7]^+$
		679	$[\text{Hf}_2\text{O}(\text{OR})_4(\text{OH})]^+$
	667	737	$[\text{Hf}_2\text{O}(\text{OR})_5]^+$
567	569	609	$[\text{Hf}_2\text{O}_2(\text{OR})_3]^+$
	404	461	$[\text{Hf}(\text{OR})_4]^+$
		453	$[\text{Hf}(\text{OR})_3(\text{OC}_3\text{H}_6)]^+$
		425	$[\text{Hf}(\text{OR})_3(\text{OCH}_2)]^+$
358		400	$[\text{Hf}(\text{OR})_3]^+$
		369	$[\text{Hf}(\text{OR})_2(\text{OEt})]^+$
		324	$[\text{Hf}(\text{OR})_2]^+$

To reviewer 2:

Comments:

In the work of Changhong Wang et al., in order to identify a volatile metallo-organic precursor for the deposition of HfO_2 films for atomic layer deposition (ALD) applications, the evaporative properties of three hafnium alkoxides were

investigated using thermogravimetric analysis (TGA). Hafnium isopropoxide, hafnium *n*-propoxide and hafnium *n*-butoxide were synthesized by electrochemical method and characterized by Fourier transform infrared spectroscopy (FT-IR), nuclear magnetic resonance (NMR) and inductively coupled plasma (ICP) analysis techniques. Synthesized samples were subjected to a simultaneous thermogravimetric–differential thermal analysis (TG–DTA) unit at 10 K min⁻¹ in a dry nitrogen atmosphere flowing at 100 mL min⁻¹. And then, a modified Langmuir equation was used to calculate vapor pressure curves for hafnium isopropoxide and hafnium *n*-propoxide. The vapor pressure curve of hafnium isopropoxide was calculated to be $\ln p = 31.157(\pm 0.200) - 13130.57(\pm 56.50)/T$. However, no curve was constructed for hafnium *n*-propoxide and hafnium *n*-butoxide because these two compounds undergo evaporation and deposition simultaneously.

The manuscript is mostly well written, the method seems to be appropriate and properly conducted. However, explanations of the experimental procedures and results are not sufficient. Some more explanations need to be added to make the study more comprehensible. In addition, the authors need some improvements to English, many sentences are absolutely not clear (see details below). The presentation of this work can be improved and have listed my suggestions below. At this stage, my recommendation for this manuscript is major revision.

Specific Comments:

Comment 1: Page 20, “Thermal property is of prime consideration.....predict the desired conditions and is favorable for process investigation.” This sentence is too complicated and not clear.

Answer 1: According to reviewer’s valuable suggestion, corresponding sentence has been revised. Thermal property is of prime consideration in evaluating the feasibility of a metallo-organic compound as a precursor in ALD, since a precursor must be thermally stable under the conditions required to transport its vapors to the substrate zone, to avoid pressure build up and escape of material. Vapor pressure (VP) is a crucial parameter of thermal property for selecting a precursor suited to ALD, because the VP is important to predict the desired conditions and is favorable for process investigation.

Comment 2: Page 21, 3.2 characterization, the scan times and resolution of the FTIR measurements should be given.

Answer 2: According to reviewer’s constructive suggestion, corresponding part

has been revised in the manuscript.

Comment 3: Page 21, “Then, the distillation temperature was raised to corresponding temperature to remove a little amount of ester.” What does the “corresponding temperature” mean? This sentence is not clear.

Answer 3: Thank you for pointing this out. Corresponding temperature of removing ester is around 423.15 K. After removal of the ester, the temperature rises again. Corresponding part has been revised in the manuscript.

Comment 4: Page 21, “And then the impurity content of hafnium oxide was converted into that of hafnium alkoxides.” I’m wondering where does the hafnium oxide comes from. More explanations should be added here.

Answer 4: According to reviewer’s constructive instruction, corresponding explanation has been added in the manuscript. In order to determine impurity contents of hafnium alkoxides, some amount of water was added into hafnium alkoxides solution for hydrolysis reaction:

After desiccation in desiccator under vacuum conditions at 100 °C for 12 h, the sample was calcined in muffle furnace at 800 °C for 2 h. The impurity contents were obtained by ICP-Mass Agilent 7500a analyzer. And then the impurity content of hafnium oxide was converted into that of hafnium alkoxides.

Comment 5: In the whole text, both “°C” and “K” were used for temperature, which should be revised to maintain consistency.

Answer 5: According to reviewer’s constructive comment, corresponding parts have been revised in the manuscript.

Comment 6: Page 21, “the direct electrochemical synthesis of hafnium alkoxides compared with the traditional method has a greater promise owing to its remarkable advantages.” “compared with the traditional method” can be deleted.

Answer 6: Thank you for pointing this out. Corresponding part has been revised in the manuscript.

Comment 7: Page 21, “Detailed reasons were described previously in our study.” “study” should be “studies”.

Answer 7: According to reviewer's comment, corresponding part has been revised in the manuscript.

Comment 8: Page 21, "hafnium alkoxides was successfully obtained by Turevskaya et al". "et al" should be "et al."

Answer 8: According to reviewer's comment, corresponding part has been revised in the manuscript.

Comment 9: Page 21, TGA, FT-IR, ¹H-NMR and ICP appeared in the main body for the first time without full name.

Answer 9: According to reviewer's valuable comment, corresponding part has been revised in the manuscript.

Comment 10: Page 22, "Peaks at around 2970 cm⁻¹ and 2860 cm⁻¹ correspond to asymmetric and symmetric stretch vibrations, respectively." Which functional group does these two peaks belong to?

Answer 10: Thank you for pointing this out. Peaks at around 2970 cm⁻¹ and 2860 cm⁻¹ correspond to $\delta(\text{C—H})$ stretching vibration.

Comment 11: Page 23, "Corresponding datum, which have been used to construct Figure 4, to calculate k are given in table 3, in which the values of p_{calc} are obtained from the fitting line of plot of p against v ." This sentence doesn't read smoothly.

Answer 11: Thank you for pointing this out. Corresponding sentence in the manuscript has been revised. Corresponding datum, which have been used to construct Figure 4, to calculate k are given in table 3. The values of p_{calc} in table 3 are obtained from the fitting line of plot of p against v .

Thanks to the referees for the thoughtful and thorough review. We have made careful modification on the original manuscript. We sincerely hope the present manuscript with the above revisions and replies is acceptable for publication in your journal.

Great thanks to you and the referees for the time and effort expended on this paper.

Best Wishes,

Sincerely yours,

Yongming Chen

Institute: Central South University

Address: No.932, Lushan Road, Changsha city, Hunan Province, China 410083

Tel: +86-18684685548

E-mail: csuchenyongming@163.com

Appendix E

Title: Determination of the vapor pressure curves and vaporization enthalpies of hafnium alkoxides using thermogravimetric analysis

Manuscript ID: RSOS-181193.R1

To editor:

Dear Editor,

Thank you very much for sending us the RSOS-181193.R1 Decision Letter on 13-Nov-2018. We really appreciate your advice and the reviewer's comments to our manuscript. We have studied all the comments carefully, and thank you very much for your time and thoughtful comments. We have now revised the manuscript in accordance with the comments and suggestions. **All changes made to the text are highlighted with a red color.** The entire comments are responded point by point as follows.

To reviewer 1:

Comments: The comments have been addressed except the following: In NMR analysis section, please add explanation of source of the impure peaks into the manuscript as the authors did in the rebuttal.

Answer: According to reviewer's constructive comment and instruction, corresponding parts have been revised in the revised manuscript.

To reviewer 2:

Comments: I have checked the revised manuscript carefully. Further discussion about the removal of impurities contents of hafnium alkoxides were added. In addition, more explanations were given which made the manuscript complete and comprehensible. However, it seems that the authors have only revised the manuscript according to part of the suggestions I have raised.

Comment 1: Although I have raised that some sentences in the manuscript are too complicated and not clear, the authors just did a few punctuation tweaks in the revised version.

Answer 1: According to reviewer's constructive comment and instruction, corresponding parts have been revised in the revised manuscript.

Comment 2: Once again, the scan times of the FTIR measurements should be given for clear presentation.

After the modification made by the authors, the revised manuscript is much better than the previous edition. However, the authors should carefully check the format and English of the article. It would be a qualified manuscript after the authors take care of these details.

Answer 2: According to reviewer's constructive comment and instruction, corresponding parts have been revised in the revised manuscript. The scan times of 32 in the FTIR measurements have been added to the revised manuscript. The format and English of the article have been carefully checked.

Thanks to the referees for the thoughtful and thorough review. We have made careful modification on the original manuscript. We sincerely hope the present manuscript with the above revisions and replies is acceptable for publication in your journal.

Great thanks to you and the referees for the time and effort expended on this paper.

Best Wishes,

Sincerely yours,

Yongming Chen

Institute: Central South University

Address: No.932, Lushan Road, Changsha city, Hunan Province, China 410083

Tel: +86-18684685548

E-mail: csuchenyongming@163.com